# Ubiquity of Emergent Hebbian Dynamics in Regularized Learning

**David Koplow** [1]   **Tomaso Poggio** [1]   **Liu Ziyin** [1,2]

## Abstract

Hebbian and anti-Hebbian plasticity are widely observed in the brain and are classically modeled as mechanistic, local homosynaptic rules stabilized by homeostatic constraints. This raises an identifiability question: does observing Hebbian/anti-Hebbian structure in synaptic updates uniquely imply an underlying Hebbian computation? We identify an alternative, emergent route. We show that near stationarity, L2 weight decay generically drives the *learning-signal* component of many update rules to align with a Hebbian direction, with alignment increasing monotonically with decay strength. This Hebbian-like signature is not specific to SGD and can arise even for non-learning or random update rules long before learning has ceased. We further show that stochastic noise in the learning signal can induce anti-Hebbian alignment, yielding a simple tradeoff with weight decay and a phase boundary in regression settings. These mechanisms do not replace standard Hebbian theory; they can coexist with genuine Hebbian plasticity and complicate the interpretation of synaptic measurements, motivating experiments that distinguish mechanistic Hebbian computation from emergent Hebbian signatures.

## 1. Introduction

A central question in computational neuroscience is the extent to which artificial intelligence systems capture the computational motifs found in the human brain. Traditionally, it has been widely believed that artificial intelligence and the human brain operate under entirely different learning mechanisms: modern AI systems rely on gradient descent for learning, whereas biological synapses are generally thought

to adapt primarily via correlation-based mechanisms, specifically Hebbian and anti-Hebbian forms of learning (Koch et al., 2013; Zenke & Gerstner, 2017; Lisman, 1989; Lamsa et al., 2007; Abbott & Nelson, 2000; Magee & Grienberger, 2020). Standard Hebbian theory views Hebbian learning as *mechanistically* implemented by local pre/post-activation homo-synaptic rules such as Spike-Time-Dependent Plasticity (STDP). This Hebbian update is widely believed to be the primary driver of learning and memory in the brain, with additional balancing mechanisms preventing runaway neuronal growth or decay (Oja, 1982; Bienenstock et al., 1982; Caporale & Dan, 2008; Froemke et al., 2005; Brzosko et al., 2019; Zenke & Gerstner, 2017). This framework has been extraordinarily successful in explaining how local plasticity can extract structure from inputs and why homeostatic constraints are essential.

At the same time, Hebbian-like signatures are often used as indirect evidence *against* global error-driven optimization in the brain, since gradient-based learning is typically viewed as requiring nonlocal error signals and coordination that neural circuits may not provide (Hebb, 2005; Rumelhart et al., 1986; Whittington & Bogacz, 2019; Lillicrap et al., 2020). This motivates a basic identifiability question: *does observing Hebbian/anti-Hebbian structure in updates uniquely imply an underlying Hebbian computation?*

In this work we show that the answer can be no. We study an emergent route to Hebbian and anti-Hebbian *signatures* that arises from the same stability considerations emphasized by standard Hebbian theory. The key idea is a balance of expansive and contractive forces: in biology, Hebbian dynamics is expansive and must be countered by homeostatic mechanisms; in machine learning, the learning signal is balanced by explicit contraction (L2 weight decay) and by stochasticity/noise. We prove and empirically demonstrate that, near stationarity, L2 weight decay generically drives the *learning-signal component* of many update rules (including SGD and even non-learning/random constructions) to align with a Hebbian direction, with alignment increasing monotonically with decay strength. Conversely, sufficiently strong noise can flip the signature, producing anti-Hebbian alignment and a simple tradeoff boundary between noise strength and weight decay (Figure 1).

Our aim is not to replace classical Hebbian theory: true Heb-

---

[1]Center for Brains, Minds and Machines, Massachusetts Institute of Technology [2]NTT Research. Correspondence to: David Koplow <dkoplow@mit.edu>.

*Proceedings of the 43rd International Conference on Machine Learning*, Seoul, South Korea. PMLR 306, 2026. Copyright 2026 by the author(s).

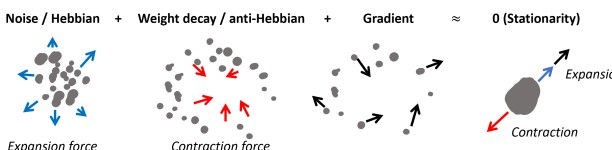

*Figure 1.* **Balance of contractive and expansive forces.** For deep learning, the noise and weight decay are, respectively, expansion and contraction forces. When they do not balance, the gradient must fill in the gap – if noise outweighs weight decay, the gradient must appear contractive; otherwise, it appears expansive. Similarly, for biology, the Hebbian dynamics is always expansive, and the anti-Hebbian dynamics is always contractive. Thus, to reach a balance, a learning signal will look like, and become aligned with, the Hebbian or anti-Hebbian rule depending on whether it is expansive or contractive.

bian processes almost certainly occur widely in the brain. Instead, we propose an additional, coexisting account of why synaptic modifications can *appear* Hebbian or anti-Hebbian: close to a homeostatic regime, regularization constraints project complex learning dynamics into Hebbian-like directions. This calls for caution in interpreting plasticity measurements and supports designing experiments that can separate genuinely Hebbian mechanisms from merely emergent Hebbian patterns.

**Contributions.** We provide:

1. A general near-stationarity mechanism: with L2 weight decay, the learning signal of broad classes of learning rules becomes positively aligned with Hebbian updates, strengthening with decay.
2. A complementary noise mechanism: strong stochastic perturbations induce anti-Hebbian alignment and yield a predictable tradeoff with weight decay.
3. Empirical validation across tasks, architectures, and learning rules, showing these signatures can appear well before learning has ceased.

This work is organized as follows. The next section discusses related works and preliminaries. Section 3 studies Hebbian signatures from regularization. Section 4 studies anti-Hebbian signatures from noise. Section 5 analyzes transient and nonstationary regimes. Additional figures are presented in the Appendix.

## 2. Related works and Background

**Hebbian Learning.** As a mathematical model, consider a hidden layer in an arbitrary network:

$$h_b = W h_a(x), \quad (1)$$

where $h_a$ is the postactivation of the previous layer, and $h_b$ is the preactivation of the current layer. In the most conventional form, the simplest homosynaptic update[1] rule states that $W$ is learned according to

$$\Delta W = s\eta h_b h_a^\top, \quad (2)$$

where $s \in \{-1, 1\}$ is the sign of learning. When $s = 1$, the rule is Hebbian, which states that if neuron $i$ causes neuron $j$ to fire, then their connection should be strengthened. Similarly, when $s = -1$, the rule is anti-Hebbian, as it tends to reduce correlation between neurons. $\eta$ is a positive time constant, which we call the "learning rate." In a neuroscience context, Eq. 2 should be regarded a discrete-time approximation to the true underlying continuous-time process, and the rule implies, for a given input,

$$\dot{W} = s\eta h_b h_a^\top = s\eta W h_a h_a^\top, \quad (3)$$

which increases the norm of $W$ when $s > 0$ and decreases it when $s < 0$. Therefore, Hebbian learning in this limit is always expansive, and anti-Hebbian learning is always contractive (see Figure 1 for a visualization). Evidence exists to show that both Hebbian and anti-Hebbian learning exist widely in the brain (Abbott & Nelson, 2000; Magee & Grienberger, 2020).

Classical formulations of Hebbian plasticity show that simple local rules can recover meaningful structure from sensory input, from normalization stabilized PCA in linear models (Oja, 1982; Sanger, 1989) to higher order feature extraction in nonlinear and Bienenstock-Cooper-Munro (BCM) style variants (Bienenstock et al., 1982; Cox & Adams, 2009). Biophysically grounded rules such as voltage-based and STDP-inspired plasticity (Clopath et al., 2010) and recurrent networks combining Hebbian excitation with anti-Hebbian inhibition (Zylberberg et al., 2011) demonstrate how decorrelation, whitening, and sparse receptive fields can arise in realistic circuits. More recent unifying work shows that many such rules can implement ICA or sparse coding objectives (Brito & Gerstner, 2016). Related models explain anti-Hebbian learning as a structured mechanism for maintaining excitation-inhibition balance and promoting decorrelation (Vogels et al., 2011; King et al., 2013). A few works have suggested that there are certain types of learning that anti-Hebbian learning can also play a role in, though this is less widely established (Zenke et al., 2017; Pehlevan et al., 2018). However, none of the prior works have suggested how the Hebbian plasticity could alternatively be an emergent phenomenon.

A key factor of our proposed mechanism is weight decay, and recent works have implied the importance of weight decay in biologically plausible learning rules (Akrout et al., 2019; Liao et al., 2024; Ziyin et al., 2025a; Bacvanski et al., 2026). The closest related work is Ziyin et al. (2025a), which suggested that weight decay could cause an alignment

---

[1]We use this term as a synonym of Hebbian learning.

to the Hebbian rule, but the mechanism is unclear, and it fails to point out the connection between noise and anti-Hebbian learning.

**Gradient Descent in the Brain.** So far, there has not been any strong evidence that the brain could implement and run any form of gradient descent, despite various theoretical proposals (Kolen & Pollack, 1994; Lillicrap et al., 2020; Whittington & Bogacz, 2019; Richards & Kording, 2023)–and observations of Hebbian plasticity are often implicitly regarded as evidence against gradient descent (e.g., see the criticism of heterosynaptic rules in (Porr & Wörgötter, 2007)). Our theory shows that gradient descent dynamics can lead to dynamics at stationarity that are consistent with the Hebbian phenomenon, and because of this, observations of Hebbian updates may be at least partially consistent with the existence of more complicated learning rules in the brain.

**Similarity between learning algorithms.** A few works are closely related to ours. (Xie & Seung, 2003) shows the equivalence of gradient descent to a form of contrastive Hebbian algorithm (CHA). However, CHA is not biologically Hebbian because it is not a homosynaptic rule, required by the Hebbian principle. There have been several other adjacent ideas to modify the Hebbian rule to lead to learning performance similar to gradient descent or even mathematical equivalences to SGD in certain types of models (Scellier et al., 2018; Xiao et al., 2019; Scellier & Bengio, 2019; Ernoult et al., 2022). But these works fail to provide a general relationship between arbitrary models trained with SGD and do not identify the key role of regularization and noise.

## 3. Learning-regularization balance produces Hebbian learning

There have been proposals that a balance between Hebbian and anti-Hebbian dynamics must happen for the brain to reach at least some form of homeostasis (stationarity) (Xie & Seung, 2003; Oja, 1982; Bienenstock et al., 1982). There is a similar effect in gradient-based training in neural networks. The use of weight decay contracts the weights to become smaller, but learning can hardly happen if the weights are too small. Therefore, any model that reaches some level of stationarity in training must have a gradient signal that is expansive and opposed to the contractive effect of weight decay.

For the layer defined in Eq. 1, the full weight update is

$$\Delta W \propto \underbrace{-(\nabla_{h_b(x)}\ell)h_a^T(x)}_{\text{learning signal}} -\gamma W, \qquad (4)$$

where $\ell$ is the loss function and $\gamma$ is the strength of weight decay.

Let us first show that close to stationarity, the raw loss gradient is contractive, and hence the negative-gradient learning signal is expansive. Close to a stationary point, the update should be small in expectation: $\mathbb{E}_x[(\nabla_{h_b(x)}\ell)h_a^T(x)]+ \gamma W \approx 0$. Thus, after right multiplying the equality by $W^T$ we substitute in Eq. 1 and find that

$$\mathbb{E}_x[(\nabla_{h_b(x)}\ell)h_b^T(x)] = -\gamma WW^\top. \qquad (5)$$

We can then use the Frobenius inner product identity to derive the sign,

$$\text{Tr}\,\mathbb{E}_x[(\nabla_{h_b(x)}\ell)\,h_b^T(x)] = \mathbb{E}_x[(\nabla_{h_b(x)}^T\ell)\,h_b(x)] \qquad (6)$$

$$= -\gamma\,\text{Tr}[WW^T] < 0. \qquad (7)$$

Therefore, on average,

$$\nabla_{h_b(x)}\ell^\top h_b(x)$$

is negative. Equivalently, the negative-gradient learning signal has a positive inner product with $h_b(x)$ on average. Now, as in (Ziyin et al., 2025b), we assume an equal-norm or concentration condition:

$$\|h_a(x)\|^2 \approx \mathbb{E}[\|h_a(x)\|^2].$$

This states that presynaptic representation norms are nearly constant across inputs. This is certainly satisfied when, for example, there is a neural collapse (Papyan et al., 2020) or when the representations are normalized. This means that the expected alignment between the learning signal and the Hebbian update is given by

$$\mathbb{E}_x\left[\left\langle -(\nabla_{h_b(x)}\ell)h_a(x)^\top, h_b(x)h_a(x)^\top\right\rangle_F\right] \qquad (8)$$

$$= -\mathbb{E}_x[\|h_a(x)\|^2\nabla_{h_b(x)}\ell^\top h_b(x)] \qquad (9)$$

$$\approx -\mathbb{E}[\|h_a\|^2]\,\mathbb{E}_x[\nabla_{h_b(x)}\ell^\top h_b(x)] \qquad (10)$$

$$= \gamma\mathbb{E}[\|h_a\|^2]\,\text{Tr}[WW^\top] > 0. \qquad (11)$$

Namely, the learning signal has positive correlation with the Hebbian update, and the alignment becomes stronger as $\gamma$ increases. See Figure 2 for an example of such alignment.

Perhaps surprisingly, a weaker form of this result generalizes to any update rule, precisely because the weight decay term always aligns with the anti-Hebbian update; at stationarity the expected learning signal must align with the Hebbian direction. Consider an arbitrary learning signal $g(x, \theta)$; the full weight update is

$$\Delta W = g(x, \theta) - \gamma W, \qquad (12)$$

where $g$ is the learning rule and $\theta$ is the entirety of all trainable (plastic) parameters. For clarity, $\eta$ is subsumed into

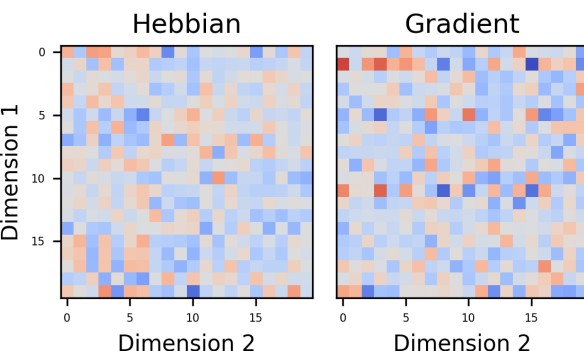

*Figure 2.* The **left** shows example weight updates with a high alignment between the learning signal $(-\nabla_W \ell)$ and the Hebbian update at the end of training with high weight decay, while the **right** image displays an example update at the end of training with no weight decay which has very low alignment. This figure shows a 20x20 subset of the direction of the Hebbian and learning signal updates for the second layer of an SCE after training with $\eta = 0.1$, and $\gamma = 0.05$, or $\gamma = 0.0$. Dimension 1 can be viewed as the output (post-synaptic) neuron (in the case of SGD, whose incoming weights we are differentiating), and Dimension 2 are input (pre-synaptic) feature/neurons that the weight projects from. We are only visualizing a 20x20 subset of these updates for clarity. Examples of low cosine similarity of the learning signal for $\gamma = 0.05$ at the start and end of training can be seen in Figure 17. In general, we find that stronger weight decay, larger learning rate, and larger batch size lead to better alignment (Figures 7 and 8).

$g$ and $\gamma$. Close to stationarity, we have that $\mathbb{E}_x[g(x, \theta)] \approx \gamma W$.

Now, consider the population cosine similarity between the learning rule and the Hebbian rule. Let

$$\bar{H}(W) := \mathbb{E}_x[h_b(x)h_a(x)^\top].$$

Then

$$\cos \theta = \frac{\langle \mathbb{E}_x[g(x,\theta)], \bar{H}(W) \rangle_F}{\|\mathbb{E}_x[g(x,\theta)]\|_F \|\bar{H}(W)\|_F}. \tag{13}$$

The direction of alignment at stationarity when $\mathbb{E}_x[g(x, \theta)] = \gamma W$ is thus

$$\langle \mathbb{E}_x[g(x,\theta)], \bar{H}(W) \rangle_F = \gamma \langle W, \mathbb{E}_x[h_b(x)h_a(x)^\top] \rangle_F \tag{14}$$

$$= \gamma \mathbb{E}_x[\|h_b(x)\|^2] > 0. \tag{15}$$

We see that the update must have a positive alignment with the Hebbian rule on average. This shows an intriguing and yet surprising fact: any algorithm with weight decay may look like a Hebbian rule, and the Hebbian rule may just be a "universal" projection of more complicated algorithms. This is a weaker but rather universal result. It is different from Eq. 11 in the sense that Eq. 11 predicts that the learning signal and the Hebbian rule are statistically correlated, whereas this result only says that they have the same direction when averaged over all stimuli. This theory can naturally be extended to the case when the weight decay strength $\gamma$ is different for different neurons, which we discuss in Appendix C.4. Also, this theory can be presented

in a fully formal style, which we present in Appendix C.5, where we also describe how this Hebbian alignment emerges outside of stationarity.

**Possible Neurobiological Mechanism.** A key feature of this simple theory is that it separately considers the effects of the learning signal and the weight decay, which, in the context of neurobiology, are likely to have different biological substrates. The learning signal is a fast process; it is likely to, for example, come from other neurons and take the form of electric currents and spikes (Lillicrap et al., 2020). The weight decay, however, is a much slower biochemical process and directly corresponds to the changes in the biochemical properties of synapses (such as a spine shrinkage (Stein et al., 2015)). Therefore, the biological realizations of these two processes are likely to take different forms and can be separately measured. This makes it particularly important to have a theory for the learning-signal component of the update, as this can be directly measured through LTD and LTP experiments of Hebbian plasticity (e.g., see (Zenke & Gerstner, 2017)). We focus on uniform L2 weight decay in this work for its ubiquity in machine learning and analytic simplicity. While Hebbian models often assume some form of L2 weight decay, the brain is unlikely to implement any perfectly uniform weight decay (Oja, 1982). Although exploring all non-uniform variants of L2 weight decay is too broad for thorough empirical testing, Appendix C.4 extends our analysis to this domain.

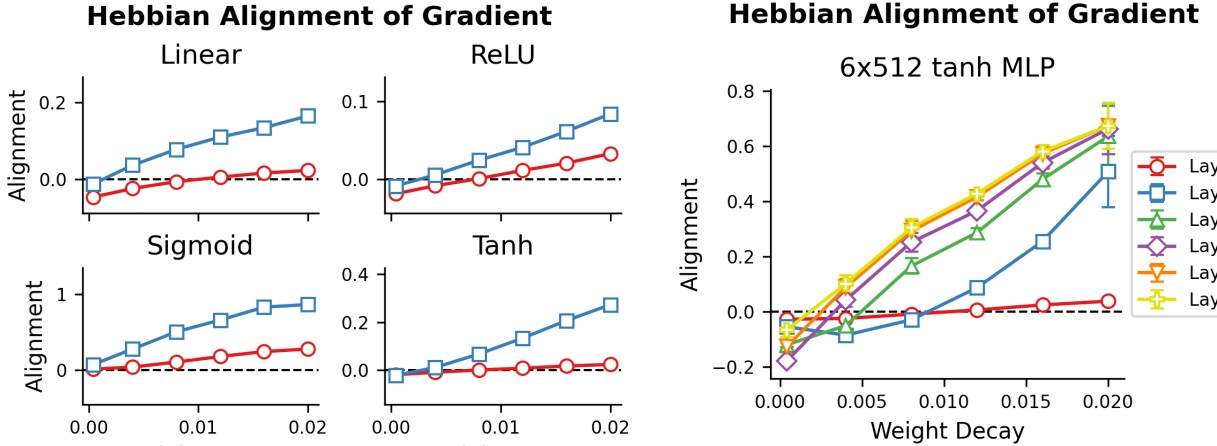

*Figure 3.* The diagram on the **left** shows that the trend of weight decay increasing Hebbian alignment of the learning signal is robust across different activation functions. The diagram on the **right** shows that the trend can generalize to deeper networks. The SCE MLPs were modified by varying the activation functions across Linear, ReLU, Sigmoid, and Tanh (**left**) and increasing the depth to 6 and layer width to dimension 512 (shown by the 6x512 tanh MLP plot on the **right**). Layer 1, Layer 2, ... Layer 6 in this diagram indicate the Hebbian alignment with the learning signal for the corresponding layer. For a small (or zero) weight decay, the learning process sometimes exhibits a weak anti-Hebbian alignment, indicated by a negative alignment with Hebbian learning. All markers represent the average alignment over the final 100 steps, averaged across 10 runs with different seeds. The error bars represent the std of the final average alignment across the seeds. The trend is very robust, and so many of the bars are obscured by the markers, particularly in the left diagram, for which the largest std was 0.012.

**Simulations.** We empirically find that this trend holds across a wide variety of different learning tasks. We ran simulations performing classification on CIFAR-10 and non-linear regression on synthetic data (Krizhevsky, 2009). We tested both MLPs and transformers, as well as a range of activation functions and optimizers. In some situations, the correlation between the two learning paradigms is very strong (e.g., in Figure 2). In our experiments, we used a default learning rate of $\eta = 0.01$ and trained for 50 epochs, which reached convergence. Since this trend only holds near stationarity–a condition achievable in full gradient descent but obscured in SGD by noise–we found it best to use larger batch sizes to compute both the gradient and Hebbian update as suggested in (Xu et al., 2023). We found a batch size of 256 to generally show Hebbian phenomena while being small enough to converge to good solutions quickly (Figure 7). To get the alignment between the updates, we compute the cosine similarity of the direction of the gradient update from the loss function (the negative gradient in SGD) and the direction of the Hebbian update. Further, most experiments reported on in this paper followed one of the following setups, and any variations will be reported when relevant.

**(1) Standard Classification Experiment (SCE):** In these experiments, we trained a small MLP with 2 layers of 128 dimensions and tanh activation using cross-entropy loss to classify CIFAR-10.

**(2) Standard Regression Experiment (SRE):** In these experiments, we trained a small MLP with two hidden layers of 128 units each and `tanh` activation, using mean squared error to predict the output of a teacher model. The teacher has the same architecture but is initialized with different random parameters. The input and output vectors are both 32-dimensional, with each element independently drawn from an isotropic Gaussian distribution. The training dataset consisted of 20,000 randomly generated training examples, and the validation dataset contained 2,000 examples. For the transformer variant of the SRE, we used a transformer with 32-dimensional token embeddings, a vocabulary size of 16, and a maximum sequence length of 32. The encoder consists of 2 layers with 4 attention heads and 32-dimensional feed-forward blocks using ReLU activations. The average of the output token embeddings is passed through the same MLP described above and compared to the teacher output.

**Classification.** We train a series of MLPs using the SCE setup to classify CIFAR-10. Figure 3 shows that as weight decay increases, so does the alignment of the learning signal with the Hebbian update. The trend persists across different activation functions. Although we still detect this trend in larger MLPS (Figure 3), we occasionally observe some layers behaving in an anti-Hebbian direction as the weight increases. Using residual connections and batch normalization can stabilize the network and make it more Hebbian; however, a deeper exploration of this quality is left for future research. We also explore the use of other regularizations in Appendix 16. By contrast, when we train the same architecture with a Hebbian rule such as Oja's on the same data, the

model performs poorly and its learning signal does not align with that of SGD at convergence (Figure 18) (Oja, 1982). This shows that classical unsupervised PCA-style Hebbian learning does not reproduce the supervised dynamics we study, and that the Hebbian-like signatures we observe are a consequence of regularized supervised optimization rather than explicit PCA feature extraction.

**Regression.** We also evaluate the generalization of this trend to student-teacher regression problems as described in SRE. We explored both MLP and Transformer models and evaluated the Hebbian alignment for learning rules outside of SGD. Recall that a key prediction of the theory is that almost any update look like a Hebbian rule when regularized. We test a variety of rules: SGD, Adam, and Direct Feedback Alignment (DFA) (Nøkland, 2016). To demonstrate that this observation is universal, we also run a setting with a randomly initialized neural network whose output is used as a learning signal, based entirely on the input data (Random NN). Notably, Random NN should not be able to *learn anything* given it is effectively only a deterministic random error signal. Results are shown in Table 1. In all cases, alignment to Hebbian learning emerges and becomes stronger as weight decay increases, regardless of the model.

## 4. Noise-Learning balance leads to Anti-Hebbian learning

We have answered the question of how Hebbian learning can be an emergent and phenomenological byproduct. The second part of the question is when we will see anti-Hebbian learning, as both Hebbian and anti-Hebbian learning are ubiquitous in the brain. Can anti-Hebbian learning also be a byproduct of more complicated learning rules?

The analysis in the previous section does not take into account the existence of noise in learning. In reality, noise is always non-negligible both in biological learning and in artificial learning. That a strong noise leads to an anti-Hebbian learning signal can already be explained by looking at a simple linear regression problem:

$$\ell(w) = \frac{1}{2}(w^T x - y)^2, \tag{16}$$

where $x \in \mathbb{R}^d$, $y \in \mathbb{R}$ are sampled from some underlying distribution at every training step. Here, we inject noise $\epsilon \in \mathcal{N}(0, \sigma I)$ to the weight before every optimization step so that $w = v + \epsilon$, where $v$ is the weight before noise injection. This could be a thermal noise that can exist ubiquitously in the brain (London et al., 2010). It can also be seen as an approximate model of the SGD noise, which causes $w$ to fluctuate around the mean (Liu et al., 2021). The learning signal and Hebbian update are

$$\Delta_{\text{SGD}} w = -x(w^T x - y), \tag{17}$$

$$\Delta_{\text{Hebb}} w = x w^T x. \tag{18}$$

The alignment between the two is

$$\mathbb{E}_\epsilon \left[ (\Delta_{\text{SGD}} w)^T (\Delta_{\text{Hebb}} w) \right] \tag{19}$$

$$= -\|x\|^2 \mathbb{E}_\epsilon \left[ (w^T x)^2 - w^T x y \right] \tag{20}$$

$$= -\|x\|^2 \left[ (v^T x)^2 + \sigma^2 \|x\|^2 - v^T x y \right], \tag{21}$$

which is negative for sufficiently large $\sigma^2$ and any $\|x\| \neq 0$. Thus, large noise leads to anti-Hebbian learning.

An interesting question is how this effect competes and trades off with weight decay. When there is a weight decay, the full weight update is $\Delta_{\text{SGD}} w = -x(w^T x - y) - \gamma w$, and so

$$\mathbb{E}_\epsilon \left[ (\Delta_{\text{SGD}} w)^T (\Delta_{\text{Hebb}} w) \right] \approx -\sigma^2 c_0 + \gamma c_1, \tag{22}$$

where $c_0$ and $c_1$ are positive coefficients that depend on the network architecture and data distribution so can be treated as constants with respect to the weight decay and noise. Thus, one expects a **phase transition** boundary at $\gamma \propto \sigma^2$. When $\gamma$ is larger than this boundary, the learning is Hebbian-like; when smaller, the learning is anti-Hebbian like. This result provides a straightforward and simple framework to potentially test and understand the Hebbian and anti-Hebbian plasticity in biology. A possible strong biological evidence that would verify this theory is the simultaneous observations of strong noise in the ambient space and anti-Hebbian plasticity. In simulation, this scaling law is verified in the experiments (Figure 4), which justifies this simple analysis.

**Simulations** We ran experiments to validate the noise prediction using a two-layer MLP with tanh activation. We used a student-teacher model to build a non-linear regression problem and trained until convergence using SGD and varying the variance of the Gaussian noise added at each training step, as well as the weight decay. There is a very smooth alignment trend with SGD, as can be seen in Figure 4. The white region shows the phase boundary between the Hebbian phase and anti-Hebbian phase, and shows a shape in accordance with the quadratic curve $\gamma \approx \sigma^2$.

We observed that at convergence, the Hebbian alignment of the learning signal is higher in low noise environments, and becomes more aligned with anti-Hebbian as the noise increases (Figure 4). Interestingly, we found that solutions with high generalization generally had low Hebbian and anti-Hebbian alignment (Figure 5).

We also observed this trend with other optimizers such as Adam (Figure 13). However, we struggled to robustly reproduce this effect outside of the last few layers of much larger networks or those doing different learning tasks, such as classification. We hypothesize this could be because the

*Table 1.* For all models, optimizers and learning rules, Hebbian alignment rises with increasing weight-decay $\gamma$. Hebbian alignment (mean $\pm$ SD, $n$ seeds = 10) at convergence is shown for the 2nd-layer gradient in a regression MLP and a sequence-to-vector transformer (1st layer for DFA). All experiments were SREs with a few modifications outside of the learning rule and weight decay specified in the table. DFA used $\eta = 0.1$ with gradient-norm $clip = 5$ and, as in the original implementation, used biases. RandomNN used gradient-norm $clip = 1$ and a target weight L2 norm of 100 to determine the sign of the update as explained in Section C.6 of the Appendix. Table elements with – indicate the model's weights collapsed to zero.

| Model | Learning Rule | Weight Decay ($\gamma$) | | | |
|---|---|---|---|---|---|
| | | 0 | $5 \times 10^{-5}$ | $5 \times 10^{-4}$ | $5 \times 10^{-3}$ |
| Regression MLP | Adam | $-0.02 \pm 0.00$ | $0.10 \pm 0.00$ | $\mathbf{0.66 \pm 0.01}$ | – |
| | SGD | $-0.10 \pm 0.01$ | $-0.06 \pm 0.01$ | $0.17 \pm 0.01$ | $\mathbf{0.59 \pm 0.01}$ |
| | DFA | $0.45 \pm 0.05$ | $0.45 \pm 0.04$ | $0.68 \pm 0.05$ | $\mathbf{0.87 \pm 0.00}$ |
| | RandomNN | $0.00 \pm 0.00$ | $0.00 \pm 0.00$ | $0.05 \pm 0.00$ | $\mathbf{0.50 \pm 0.00}$ |
| Transformer | Adam | $-0.02 \pm 0.02$ | $0.50 \pm 0.24$ | $\mathbf{0.99 \pm 0.02}$ | – |
| | SGD | $0.00 \pm 0.01$ | $0.04 \pm 0.01$ | $0.47 \pm 0.06$ | $\mathbf{0.88 \pm 0.03}$ |
| | DFA | $0.08 \pm 0.03$ | $0.07 \pm 0.02$ | $0.11 \pm 0.02$ | $\mathbf{0.12 \pm 0.02}$ |
| | RandomNN | $0.00 \pm 0.00$ | $0.00 \pm 0.00$ | $0.01 \pm 0.00$ | $\mathbf{0.09 \pm 0.01}$ |

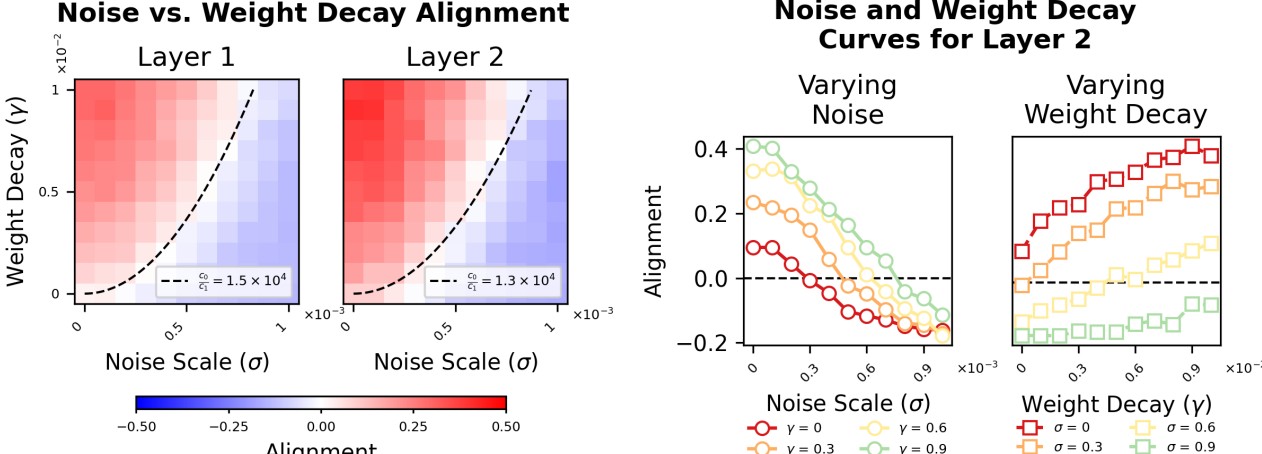

*Figure 4.* As the noise increases, the Hebbian alignment decreases and higher weight decays lead to higher Hebbian alignment (**right**). The figure on the **left** displays a heatmap of the Hebbian alignment of the learning signal at convergence for a number of different additive noises and weight decays; there is a clear quadratic curve at zero-alignment as predicted by the theory. The SRE was augmented by adding noise to each parameter at the start of each iteration with a mean of zero and the specified standard deviation on the diagram. The trend is even clearer when we follow the behavior of varying the noise of a specific weight decay (Varying Noise) or the weight decay of a specific noise standard deviation (Varying Weight Decay). Each cell on the left and marker on the right represents a single run.

magnitude of the weights does not have as much of an effect on the quality of the learned representations in larger non-linear networks, so the gradient signal does not necessarily need to point in a direction that contracts weights. We also find that adding other types of biologically plausible constraints during learning, such as a sparsification term on layer activations, can lead to a stronger anti-Hebbian alignment of the gradient.

## 5. Transient phases of Hebbian and Anti-Hebbian learning

As we mentioned in Section 3, the results are applicable when the dynamics are not yet fully stationary. While the argument we had suggested that one would only observe the Hebbian alignment close to convergence, our empirical results suggest that the alignment is present for much of training. Two key phenomena we discover are the initial alignment bump and the steady state Hebbian oscillations. Outside stationarity, the learning signal often dominates regularization. So it is sufficient to consider the full weight update directly.

## Generalization Trends

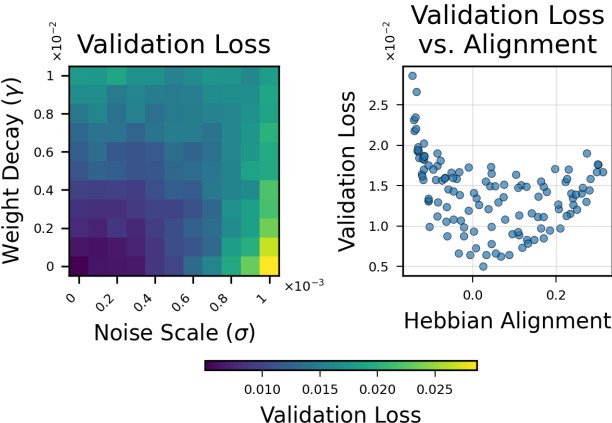

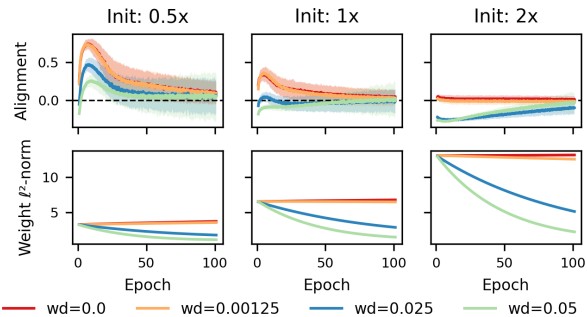

*Figure 5.* Best performance of the model is achieved when it is not Hebbian or anti-Hebbian on average. The **left** image displays the student validation loss for the experiment in Figure 4, while the **right** image shows a scatter plot of the validation loss vs. Hebbian alignment of the gradient. There seems to be some weak saddle phenomena in loss that occur at the phase transition boundary of Hebbian alignment with respect to noise and scale. The validation loss reduces as both weight decay and noise get smaller. Each cell on the left, and circle on the right, represents a single seed.

**Initial Hebbian alignment bump** Particularly for networks with ReLU activations, there is a bump in Hebbian alignment of the learning signal that appears to be strongly dependent on initialization scale and learning rate at the beginning of training (Figure 6). During this initial phase of alignment, the full weight update of SGD also increases in alignment to a pure Hebbian update. This early stage of alignment seems to be the result of general feature learning, as the actual scale of weight norms does not change substantially at the start of this period, and with positive weight decays decreases. A higher learning rate makes this process happen faster.

When we examine the Hebbian alignment of the weight updates for individual neurons in the model, a striking pattern appears. During this period, individual neurons seem to take on Hebbian or anti-Hebbian learning roles that can persist for many steps (see Figure 20). Like the average behavior of the model, the length of these phases increases as the weight initialization scale magnitude and learning rate decrease. The ratio of anti-Hebbian to Hebbian neurons increases with weight decay.

**Hebbian and Anti-Hebbian steady state oscillations** There is a phase change that occurs after the initial coherent phase of learning, which is accompanied by a strong shift in the magnitude of the alignment intensity. After this phase change, individual neurons often, though not always, seem to oscillate between strongly Hebbian or anti-Hebbian weight updates (Figure 19). Since near stationarity, the magnitude of the parameters should not increase or decrease on

*Figure 6.* For some activations at low learning rates, there is a sharp jump in Hebbian alignment of the weight update when training with SGD; the size of this jump depends on initial conditions. During this phase, the weight norm decreases monotonically, suggesting the effect is due to feature alignment rather than parameter scale. This experiment used a SCE with $\eta = 0.001$. Init: 0.5x, 1x, and 2x indicate the constant that is used to scale the default torch initialization. The plots above show single seeds to better demonstrate the evolution over time, but the trend is persistent across different seeds.

average, we also find the mean of the full weight updates to converge to zero (Figure 15). Often, we find that models with better generalization exhibit strong Hebbian/anti-Hebbian oscillations; however, strong oscillations do not necessarily entail strong generalization.

## 6. Discussion

This study suggests that Hebbian and anti-Hebbian plasticity can be understood as emergent regimes of gradient-based optimization in addition to their traditional roles as distinct learning principles. By analyzing the interaction between stochastic gradient descent, weight decay, and stochastic perturbations, we demonstrated that the expected gradient update direction aligns with classic Hebbian plasticity when contraction due to regularization dominates, and switches to an anti-Hebbian alignment when expansion driven by noise prevails. The resulting phase boundary satisfies a simple scaling relation, and the phenomenon was observed across a broad spectrum of architectures, objectives, and alternative update rules.

Our results have two primary implications. First, our results show that Hebbian and anti-Hebbian plasticity can emerge as regimes of gradient-based optimization, in addition to their conventional role as fundamental learning mechanisms. Second, the presence of Hebbian or anti-Hebbian signatures in neurophysiological data need not be interpreted as evidence against global error-driven optimization in the brain; such local plasticity patterns may arise as epiphenomena of an underlying optimization process. There are thus two

alternative routes of Hebbian plasticity:

1. **Mechanistic Hebbian**: Classic fire together, wire together, perhaps with modulation;
2. **Emergent Hebbian**: A regularized learning rule close to stationarity.

From the neuroscience side, an interesting and important problem would be to develop ways to distinguish these two mechanisms. From the machine learning side, an interesting question would be to identify interesting implications of such emergent behaviors. For example, recent works showed that the biological representations can be quite well aligned with the latent representations of artificial neural networks (Yamins & DiCarlo, 2016), which lends further support to the idea that they might learn with similar mechanisms.

There are a few limitations that provide interesting areas for future research. We only dealt with smaller models in our experiments. While this decision was reasonable given the scope of this paper, it leaves open the question of whether or not we see Hebbian dynamics in much larger-scale models. As mentioned in the text, as we expanded our models and used different optimizers, we often saw strong average anti-Hebbian alignment of a subset or all of the layers, even at high weight decays. This likely results from our stationarity condition not holding in these models. However, we do not yet have a theory for why and when these regions of anti-Hebbian alignment occur.

## Impact Statement

This work is theoretical and explanatory in nature, and we do not anticipate any direct negative societal consequences stemming from its findings. Our results elucidate how Hebbian and anti-Hebbian plasticity signatures can arise from regularized and noisy learning dynamics without requiring actual computational Hebbian mechanisms. This perspective may positively influence both neuroscience and machine learning by refining how experimental data are interpreted and encouraging more rigorous experimental designs. In machine learning, our analysis may help in understanding optimization dynamics and a surprising link to the human brain. However, it does not propose anything that would raise issues related to misuse, fairness, privacy, or environmental impact.

## Acknowledgments

Thanks to Professor Tomaso Poggio for providing invaluable feedback and discussions on this work. Compute was provided by MIT's Engaging Cluster. D.K. was supported by the Center for Brains, Minds and Machines NSF grant CCF-1231216.

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

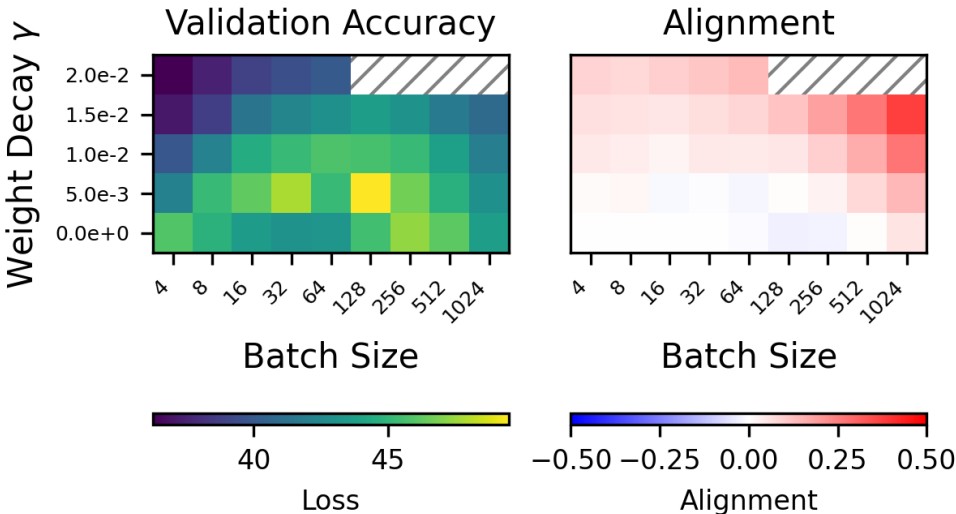

*Figure 7.* The optimal performance seems to be at a critical position between Hebbian and anti-Hebbian gradient alignment when varying batch size and weight decay. This shows the accuracy (**left**) and the Hebbian alignment of gradient update (**right**) for SCEs with a variety of weight decays and batch sizes. The striped background indicates NaN values.

## A. Reproduction

All experiments were run on MIT's OpenMind cluster using Quadro RTX 6000 GPUs and cumulatively took under 50 hours of compute time.

## B. LLM Usage

The authors used LLMs to assist in editing the manuscript and writing experimental code.

## C. Experiments

In the following document, we provide additional figures and explanations that were referenced in the main text.

### C.1. Additional influences on Hebbian alignment and generalization

C.1.1. BATCH SIZE

See Figure 7.

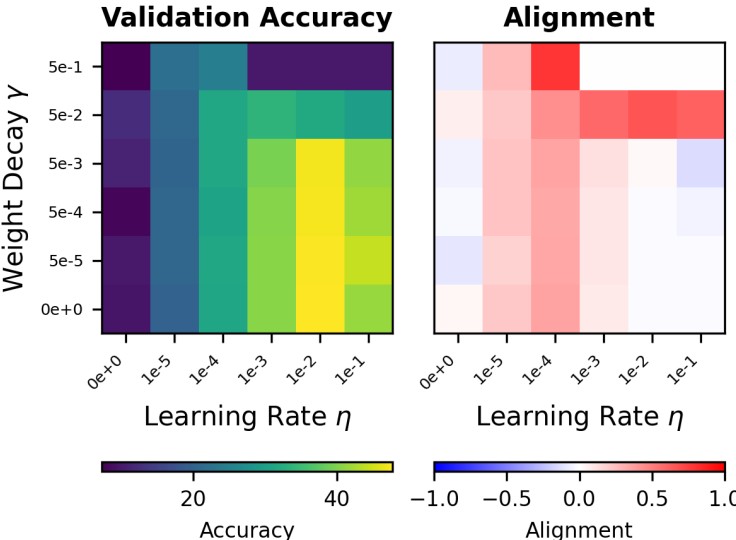

*Figure 8.* The optimal performance seems to be at a critical position between Hebbian and anti-Hebbian gradient alignment when varying learning weight and weight decay. This shows the accuracy (**left**) and the Hebbian alignment of gradient update (**right**) for SCEs with a variety of weight decays and learning rates.

### C.1.2. LEARNING RATE

See Figure 8.

### C.1.3. MODEL SCALE

See Figure 9.

$$
h = \begin{cases}
\dfrac{-(i+o) + \sqrt{(i+o)^2 + 4t}}{2}, & t = 10^6 \\[2ex]
\dfrac{-(i+o) + \sqrt{(i+o)^2 + 20t}}{10}, & t = 10^7 \\[2ex]
\dfrac{-(i+o) + \sqrt{(i+o)^2 + 28t}}{14}, & t = 5 \cdot 10^7
\end{cases}
\tag{23}
$$

Where $i$ is the dimension of the input, $o$ is the dimension of the output and $t$ is the target parameter count.

### C.1.4. MODEL SPARSITY

See Figures 10 and 11.

### C.1.5. FROZEN PARAMATERS

See Figure 12

### C.1.6. TRAINING DURATION

### C.1.7. NOISE

See Figure 13 and 14.

## Model Size vs Weight Decay: Hebbian Alignment

*Figure 9.* This diagram shows the effect of model size on Hebbian alignment and weight decay. Each point represents the mean of the alignment for the final 200 steps of the run ± the std across 10 seeds. The trend of weight decay leading to increased Hebbian alignment of the learning signal holds with larger models as well. The diagram above shows the alignment of the second layer of the respective MLPs. The MLPs had the following number of total layers: 3, 7, and 9 for the 1M, 10M, and 50M models, respectively. All hidden dimensions were assigned to reach the target parameter count as closely as possible. See equation 23 for how the exact hidden dimensions were computed. The trend is very robust so a number of the error bars are obscured by the markers.

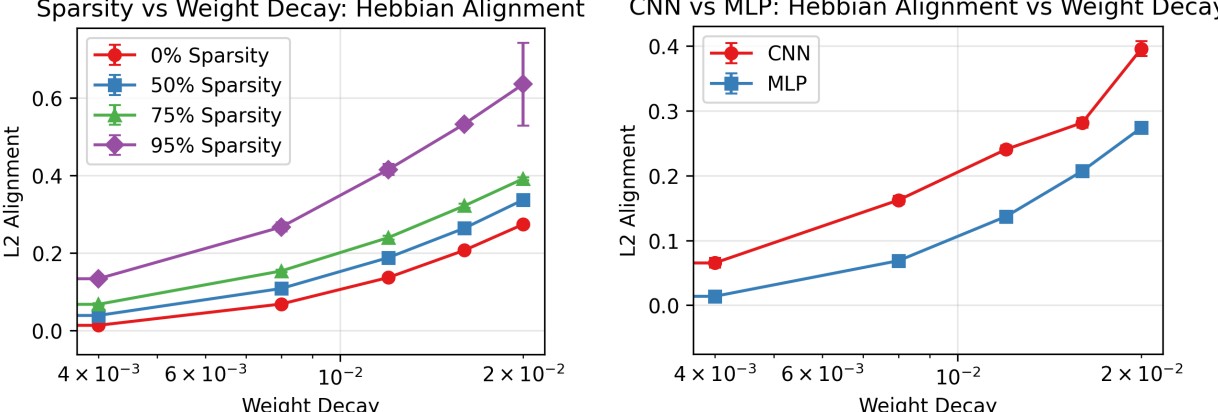

*Figure 10.* Empirically, the Hebbian alignment of the learning signal increases with sparsity **(left)**. We also see that the linear layers in a convolutional neural network, which are highly sparse, have increased Hebbian alignment. Each point represents the mean of the alignment of the second MLP layer for the final 200 steps of the run ± the std across 10 seeds. The models on the left were the standard MLPs with varying sparsity. The MLP on the right was a standard MLP and the CNN had the following architecture: Conv Layers: $(c_{in} = 3, c_{out} = 32, s = 3, p = 1)$, $(32, 64, 3, 1)$, $(64, 128, 3, 1)$ MaxPool: $(2, 2)$ Linear hidden dimensions: 2048, 512, 256. The trend is very robust so a number of the error bars are obscured by the markers.

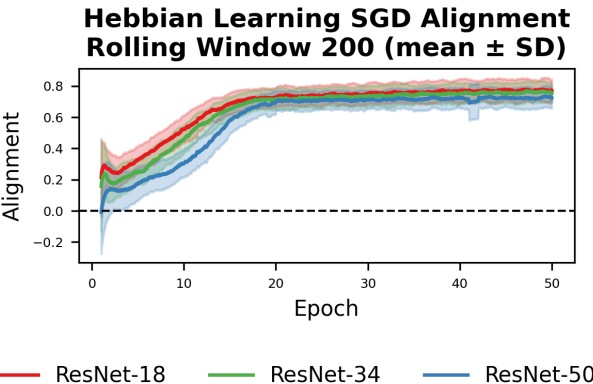

*Figure 11.* This figure shows an example SCE run with an identical training and model setup as the convolutional network described in Figure 10 but using different ResNet models as the backbone instead.

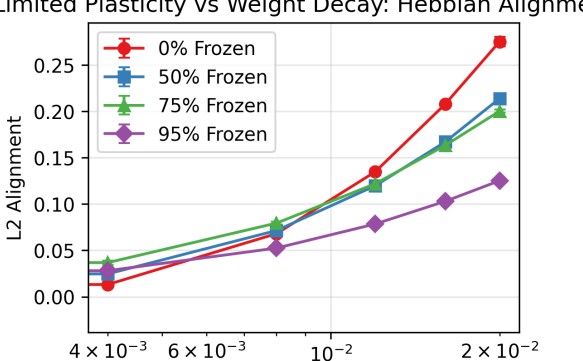

*Figure 12.* The alignment decreases as the fraction of parameters of the standard MLP that are frozen increases; however, the trend still persists. Each point represents the mean of the alignment for the final 200 steps of the run ± the std across 10 seeds. The trend is very robust so a number of the error bars are obscured by the markers.

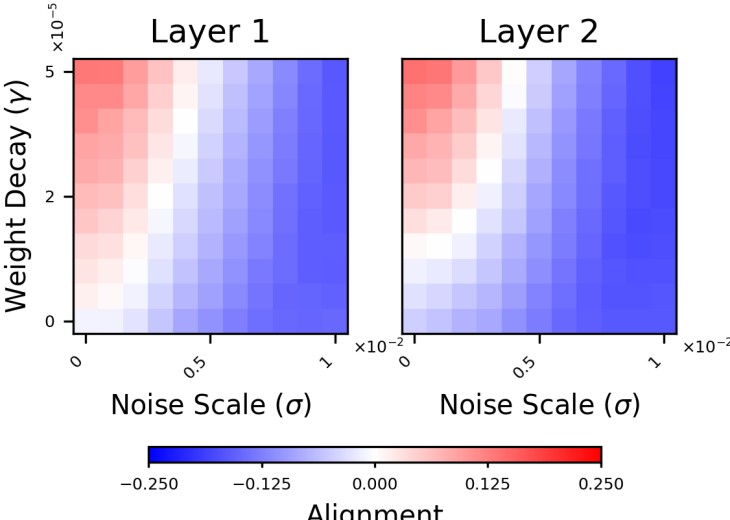

*Figure 13.* Again, there is a clear trend that even for the Adam optimizer, as noise increases, alignment of the learning signal decreases, and as weight decay increases, so too does alignment. Adam was very sensitive to the parameter ranges for which we'd see the trend, so we used a different weight decay and standard deviation range than the prior experiment. However, the rest of the architecture and experimental setup are identical to that described in Figure 4.

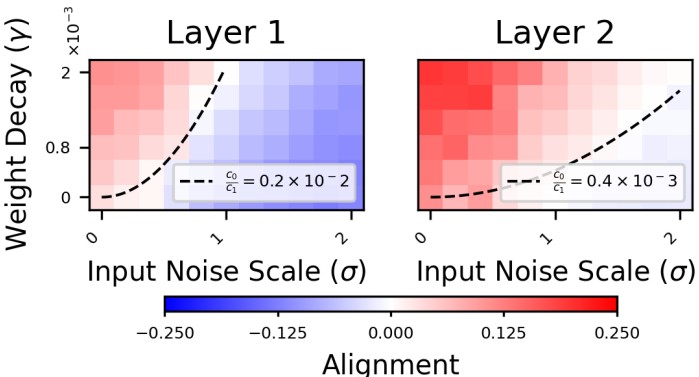

*Figure 14.* Additive noise to the input can also lead to anti-Hebbian learning. Since noise is only added to the input of the network, the exact phase boundary changes with depth. The results depicted above are from a SRE with input noise injected.

### C.1.8. Full update vs. learning signal

As defined in the terminology section, the learning signal $g(x, \theta) \equiv -\nabla_W \ell(\cdot)$ represents the gradient contribution to weight updates, while the full weight update $\Delta W = \eta(g(x, \theta) - \gamma W)$ includes both the learning signal and regularization terms. While we see that the learning signal aligns on average with the Hebbian update, the full weight update can not, otherwise the weight would explode. See Figure 15 for a visualization of the alignment of the learning signal and the full weight update over the course of training. Still, we see a very interesting trend where the full weight update often has strongly Hebbian or anti-Hebbian updates that, on average, cancel.

### C.1.9. Other regularization techniques

See Figure 16.

## Hebbian Alignment and Loss of Learning Signal
## and Full Weight Update During Training

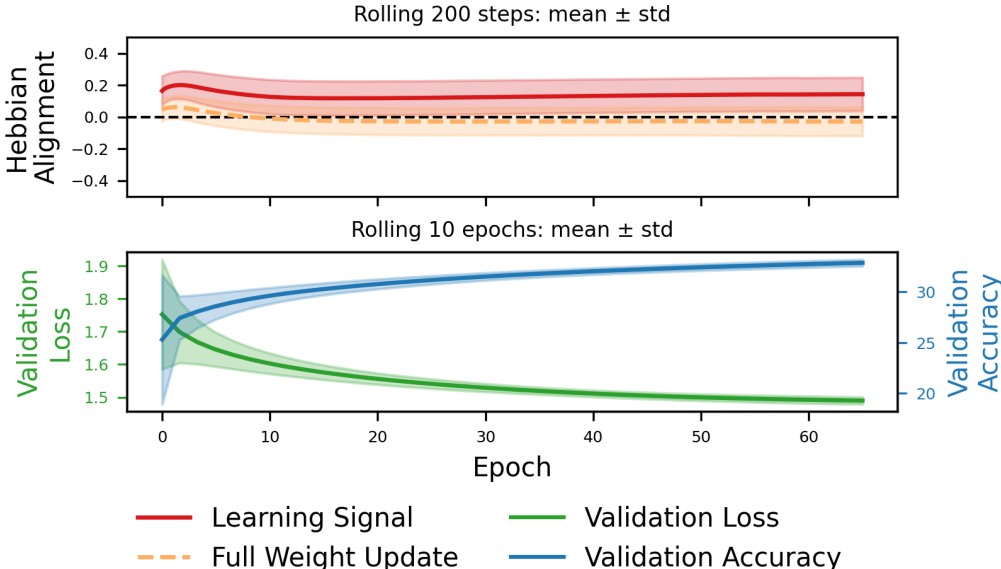

*Figure 15.* Comparison of Hebbian alignment for learning signal vs. full weight update during training (**top**) and the corresponding validation loss and accuracy (**bottom**). The learning signal alignment shows the characteristic patterns described in the main text, while the full weight update alignment approaches zero near stationarity as expected, since the mean of the full update must be zero at convergence. Individual neuron-wise signals can still oscillate between Hebbian and anti-Hebbian phases even when the mean full update is zero. Learning Signal alignment begins and persists far before learning has stopped. The Hebbian alignment is relatively low given the the low base weight decay used for SCE but reaches a non-trivial positive alignment long before convergence and while useful features are still being learned. The above graph is sampled from a single run to show the evolution over time of the alignment, though this trend is very consistent.

### C.2. Example alignments during training

C.2.1. Low alignment update at end of training

See Figure 17.

### C.3. Hebbian learning does not lead to gradient alignment

See Figure 18.

C.3.1. Full Weight Update Hebbian Oscillation

See Figures 19 and 20.

### C.4. Non-Uniform L2 Weight Decay

In the main text, we discussed for notational simplicity the case when the weight decay is uniform across all neurons. The argument can be similarly and simply extended to the case where different neurons have different rates of weight decay. We tackle that situation here.

Let $W_{i:}$ denote the $i$-th row of the weight matrix. Interpreting $i$ as the index of the neuron, this row can be seen as the synaptic efficacies of the synapses of this neuron. Here, we will allow every neuron to have a different weight decay. This is

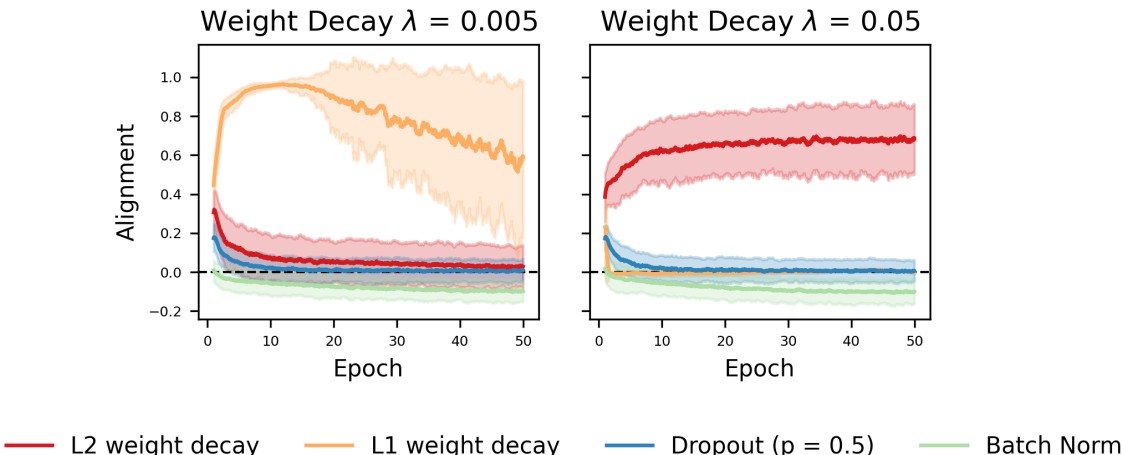

*Figure 16.* Other regularization techniques have a variety of effects on the Hebbian alignment of the learning signal. While we only developed a theory for L2 weight decay, the alignment seems to exist for some other regularizers as well when used to augment SCEs. Batch normalization seems to have a modest but persistent anti-Hebbian effect, while both L1 and L2 weight decay can have a Hebbian effect, and Dropout has no effect. While the trends above do seem robust across other seeds, the plots above show the evolution of single seeds over time to better visualize the evolution of the alignment throughout training. We present these qualitative findings for completeness; a deeper analysis of other regularization techniques is outside the scope of this work.

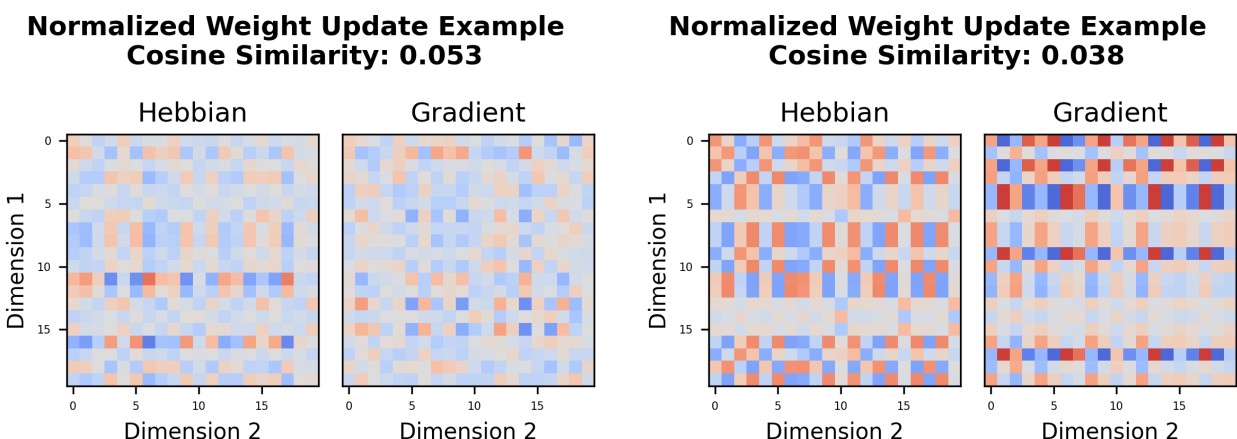

*Figure 17.* With weight decay, even after the first epoch (**left**), there starts to be an alignment of the directions; at convergence (**right**), even when specific steps have low cosine similarity, there is still clearly a lot of similar structure. At the end of training, many learning signals with low Hebbian alignment still share a surprising amount of structure. The plots above are from a SCE with $\eta = 0.1$ and $\gamma = 0.05$.

equivalent to the following generalized form of weight decay:

$$\frac{\gamma}{2} \operatorname{Tr}[W^T D W], \tag{24}$$

where $D$ is a diagonal positive-definite (PD) matrix. $D_{ii}$ is exactly the rate of decay for all the synapses of the $i$-th neuron. Of course, mathematically, this can be generalized a little bit further to allow different neurons to have correlated rates of decay, which could be biologically reasonable if these neurons are close in location. To achieve this, one simply has to allow $D$ to be a generic PD matrix, which includes the diagonal case as a special case.

For a generic learning rule $g(x, \theta)$ given in (12), the corresponding learning dynamics is:

$$\Delta W = g(x, \theta) - \gamma D W, \tag{25}$$

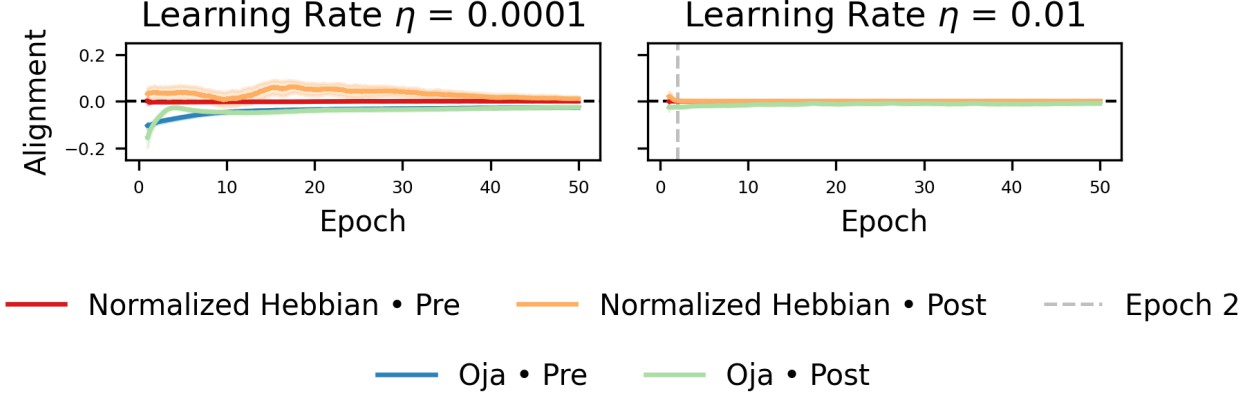

*Figure 18.* No standard interpretation of Hebbian learning produces alignment with SGD at convergence. The plots above show a different learning setup than the standard SCE; rather than training with SGD and computing the alignment of the learning signal with a Hebbian update at each update step, the model is trained with various common interpretations of the Hebbian learning objective, then this update is compared to the supervised loss gradient at each step computed with back propagation. The graph shows the mean SGD alignment of the second layer's updates, ± the standard deviation over a 200-iteration window, when trained with various versions of the Hebbian learning rule for two different learning rates. While we found the above trends to hold robustly across various seeds, each line represents only a single run smoothed over time to better demonstrate the evolution of the learning rules with respect to time. The *Normalized Hebbian* learning rule is the generic Hebbian algorithm with weight standardization after every step. The second algorithm is Oja's rule. We also tested the pre-activation and post-activation versions of both. Initially, there is either a very small positive or negative alignment between the Hebbian direction and the gradient, but the average alignment of every combination approaches zero at steady state. We include the dashed reference in the higher learning rate plot to provide intuition for how quickly updates accumulate under different learning rates. Though the runs are not directly comparable, the dashed line marks the point where the cumulative update magnitude is roughly four times that of the lower learning rate run. Since the learning rates differ by a factor of 100, this corresponds to (1/25) of the total training epochs for the higher learning rate.

where $g$ is the learning rule and $\theta$ is the entirety of all trainable (plastic) parameters. For clarity, $\eta$ is subsumed into $g$ and $\gamma$. Close to stationarity, we have that $\mathbb{E}_x[g(x, \theta)] \approx \gamma DW$.

The direction of alignment at stationarity when $\mathbb{E}_x[g(x, \theta)] = \gamma DW$ is thus

$$\left\langle \mathbb{E}_x[g(x, \theta)], \bar{H}(W) \right\rangle_F = \gamma \left\langle DW, \mathbb{E}_x[h_b(x)h_a(x)^\top] \right\rangle_F \tag{26}$$

$$= \gamma \mathbb{E}_x[h_b(x)^\top D h_b(x)] > 0, \tag{27}$$

where $\|h_b\|_D^2 = h_b^T D h_b$. We are done. Therefore, the theory extends naturally to the case when there is a nonuniform weight decay rate across neurons.

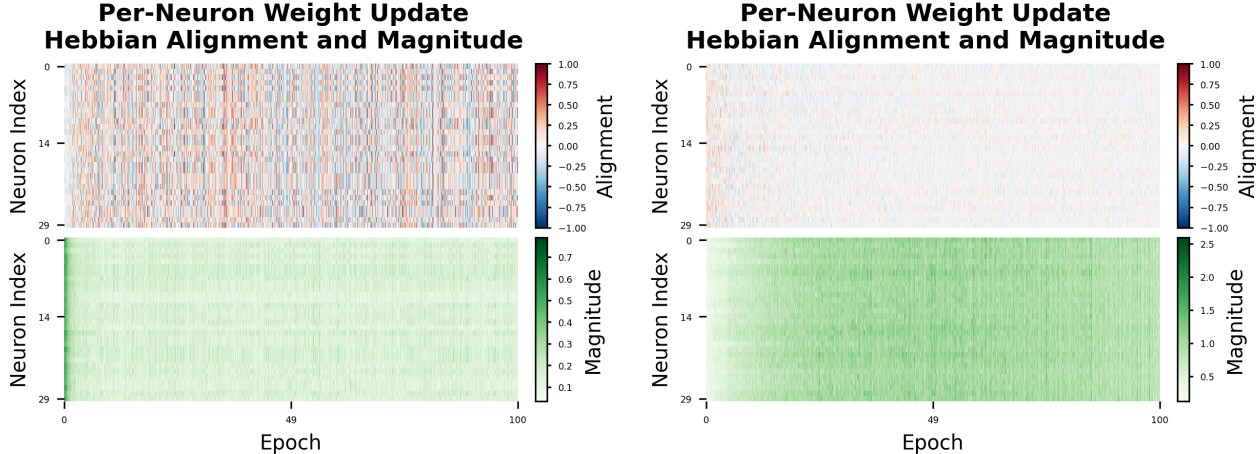

*Figure 19.* The full weight update of neurons in the neural network strongly oscillates between positively and negatively aligned to the Hebbian update late into training with weight decay (**left**). As can be seen by the blue and red stripes, there is some form of global coherent oscillation in alignment at higher weight decays. This phenomenon becomes much weaker without weight decay (**right**). Both diagrams show 30 example neurons from the second hidden layer of an MLP trained on the standard regression experiment over the course of training. The diagram on the left has a $\gamma = 0.05$ while the one on the right has a gamma of 0.0, both used tanh activations. Our experiments suggest that the oscillation in alignment for individual neurons is not related to the magnitude of the weight update that neuron is receiving, though the experiment run without weight decay does have higher magnitude weight updates since the weights are larger.

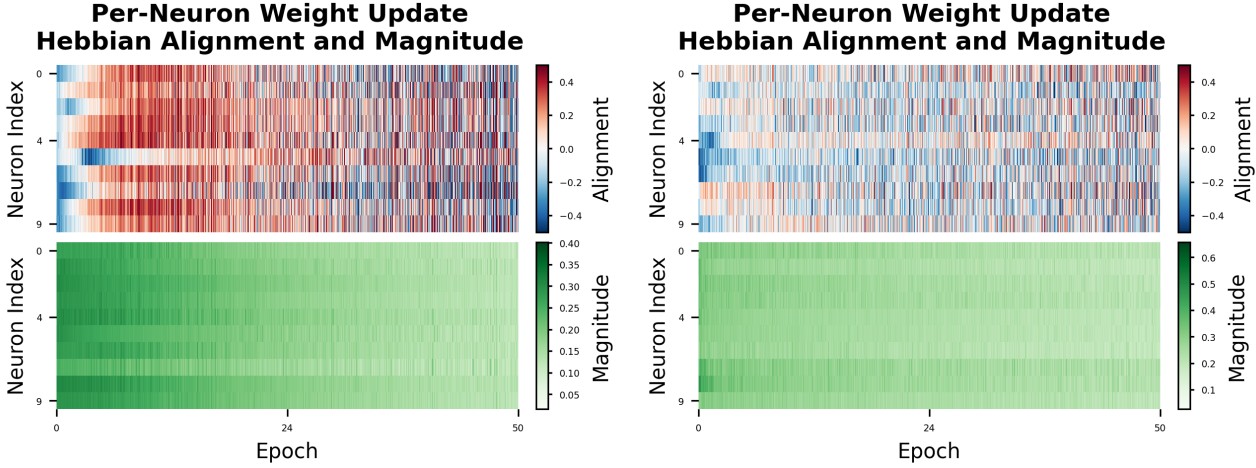

*Figure 20.* Early into training neurons often take on roles where their updates strongly bias towards Hebbian or Anti-Hebbian alignment. The diagram on the **left** depicts a SCE with ReLU activations, a 0.5x initialization scaling, a learning rate of 0.001 and a weight decay of 0.05. The diagram on the **right** depicts a similar SCE with ReLU activations and a learning rate of 0.01, but a lower weight decay of 0.025 and a default initialization.

## C.5. Formal Theory

In this section, we generalize Hebbian alignment outside of exact stationarity.

### C.5.1. THE NETWORK

Consider a single layer within a neural network defined by the weight matrix $W \in \mathbb{R}^{n_b \times n_a}$. When an input $x$ passes through the network, it triggers a presynaptic activity vector $h_a(x) \in \mathbb{R}^{n_a}$, which in turn produces the postsynaptic preactivation vector $h_b(x; W) = W h_a(x) \in \mathbb{R}^{n_b}$.

During training, the layer receives a learning-signal contribution to its update, denoted as $g(x; W) \in \mathbb{R}^{n_b \times n_a}$. Under standard gradient descent, this signal is simply the negative gradient of the unregularized loss on that example, $g(x; W) = -\nabla_W \ell(x; W)$. Throughout this analysis, we assume that both the learning signal and the presynaptic energy are well-behaved bounded expectations:

$$\mathbb{E}_x \|g(x; W)\|_F < \infty, \qquad \mathbb{E}_x \|h_a(x)\|^2 < \infty.$$

### C.5.2. DEFINING DISTANCE FROM STATIONARITY

We analyze the average behavior across the dataset of the learning signal, $G(W) := \mathbb{E}_x[g(x; W)]$. If we introduce L2 weight decay with a strength of $\gamma > 0$, the overall population update direction for this layer becomes $G(W) - \gamma W$.

This gives us a natural baseline for equilibrium. We say a layer has reached **population balance** (exact stationarity) when this total update is zero, meaning $G(W) = \gamma W$. To track how far the layer is from this ideal state, we define a distance metric:

$$d_{\text{stat}}(W) := \|G(W) - \gamma W\|_F.$$

We can define an analogous population update for the Hebbian rule. To begin, the Hebbian update for a single $x$ is defined as $H(x; W) = W h_a(x) h_a(x)^\top$. Aggregating this over the population yields the expected Hebbian update:

$$\bar{H}(W) := \mathbb{E}_x[H(x; W)].$$

To quantify the alignment between the true learning signal and the Hebbian update, we use the **population Frobenius alignment**, $A_F(W) := \langle G(W), \bar{H}(W) \rangle_F$, where $\langle U, V \rangle_F := \text{Tr}(U^\top V)$ denotes the standard Frobenius inner product. This metric measures the directional consistency between the expected network gradient and the Hebbian trajectory; hence, positivity ($A_F(W) > 0$) implies that the network dynamics are Hebbian aligned.

We also need to define another helpful quantity which we will call the **Hebbian margin**, which is the alignment between the Hebbian direction and model's weights $M_F(W) := \langle W, \bar{H}(W) \rangle_F$. By utilizing the linearity of expectation and the properties of the inner product, we can expand this margin:

$$\begin{aligned}
M_F(W) &= \left\langle W, \mathbb{E}_x[W h_a(x) h_a(x)^\top] \right\rangle_F \\
&= \mathbb{E}_x\left[\left\langle W, W h_a(x) h_a(x)^\top \right\rangle_F\right] \\
&= \mathbb{E}_x\left[\|W h_a(x)\|^2\right].
\end{aligned}$$

Because this simplifies to an expected squared norm, the margin $M_F(W)$ is guaranteed to be positive whenever the layer has a nonzero preactivation with any positive probability.

### C.5.3. ANALYZING ALIGNMENT AS A FUNCTION OF DISTANCE

With these definitions established, we can now mathematically understand how does distance from stationarity degrade alignment?

We start by breaking the population learning signal into two components:

$$G(W) = \gamma W + \big(G(W) - \gamma W\big).$$

If we take the Frobenius inner product of this expression with the population Hebbian update $\bar{H}(W)$, linearity allows us to split the alignment into two distinct parts:

$$\begin{aligned}
A_F(W) &= \langle G(W), \bar{H}(W) \rangle_F \\
&= \gamma \langle W, \bar{H}(W) \rangle_F + \langle G(W) - \gamma W, \bar{H}(W) \rangle_F.
\end{aligned}$$

The first term is our positive Hebbian margin, scaled by weight decay ($\gamma M_F(W)$). For the second term, we apply the Cauchy–Schwarz inequality to find its worst-case behavior, yielding:

$$\langle G(W) - \gamma W, \bar{H}(W) \rangle_F \geq -\|G(W) - \gamma W\|_F \|\bar{H}(W)\|_F.$$

Substituting this and our metric $d_{\mathrm{stat}}(W)$ back into the equation gives us a clean lower bound for alignment:

$$A_F(W) \geq \gamma M_F(W) - d_{\mathrm{stat}}(W)\|\bar{H}(W)\|_F = \gamma \mathbb{E}_x[\|W h_a(x)\|^2] - d_{\mathrm{stat}}(W)\|\bar{H}(W)\|_F.$$

This inequality reveals the exact boundary conditions of our system. If the layer is perfectly balanced ($d_{\mathrm{stat}}(W) = 0$), the right-hand term vanishes, and we are left with a purely positive alignment. However, it also proves that exact stationarity is *not* a strict requirement. As long as the Hebbian update is nonzero and the distance from stationarity remains small enough to satisfy the bound:

$$d_{\mathrm{stat}}(W) < \frac{\gamma M_F(W)}{\|\bar{H}(W)\|_F},$$

the Frobenius alignment $A_F(W)$ is guaranteed to remain positive ($A_F(W) > 0$) and degrade linearly with distance from population balance.

### C.5.4. COSINE SIMILARITY

While the Frobenius alignment establishes the *direction* of positivity, evaluating it as an angle provides a scale-invariant perspective. Assuming non-trivial conditions ($W \neq 0$, $G(W) \neq 0$, $\bar{H}(W) \neq 0$), we define the **population cosine alignment** as:

$$A_{\cos}(W) := \frac{\langle G(W), \bar{H}(W) \rangle_F}{\|G(W)\|_F \|\bar{H}(W)\|_F}.$$

We also define a reference metric, the baseline cosine alignment at exact balance:

$$A_{\cos}^0(W) := \frac{\langle \gamma W, \bar{H}(W) \rangle_F}{\|\gamma W\|_F \|\bar{H}(W)\|_F} = \frac{\langle W, \bar{H}(W) \rangle_F}{\|W\|_F \|\bar{H}(W)\|_F}.$$

Because the numerator here is simply our nonnegative Hebbian margin $M_F(W)$, this reference baseline $A_{\cos}^0(W)$ is intrinsically positive.

To bound our actual cosine alignment, we revisit our earlier Frobenius bound. By substituting the relation $M_F(W) = A_{\cos}^0(W)\|W\|_F\|\bar{H}(W)\|_F$ into that bound, we can factor out the Hebbian norm to isolate the impact of our distance from stationarity:

$$\langle G(W), \bar{H}(W) \rangle_F \geq \|\bar{H}(W)\|_F \left[ \gamma A_{\cos}^0(W)\|W\|_F - d_{\mathrm{stat}}(W) \right].$$

Simultaneously, the triangle inequality allows us to bound the total magnitude of the learning signal in the denominator, giving us:

$$\|G(W)\|_F \leq \gamma\|W\|_F + d_{\mathrm{stat}}(W).$$

We can integrate these two pieces by defining the **relative distance from balance** as $q(W) := \frac{d_{\mathrm{stat}}(W)}{\gamma\|W\|_F}$. If we assume that this relative distance is smaller than our reference cosine ($q(W) < A_{\cos}^0(W)$), the numerator of our cosine alignment remains strictly positive. Combining our numerator and denominator bounds yields:

$$A_{\cos}(W) \geq \frac{\gamma A_{\cos}^0(W)\|W\|_F - d_{\mathrm{stat}}(W)}{\gamma\|W\|_F + d_{\mathrm{stat}}(W)}$$
$$= \frac{A_{\cos}^0(W) - q(W)}{1 + q(W)} > 0.$$

This demonstrates that the cosine alignment stays positive as long as the relative deviation from balance does not outpace the baseline Hebbian cosine.

Finally, we can simplify this fractional bound into an intuitive linear relationship. Through basic algebraic rearrangement, we note that:

$$\frac{A_{\cos}^0(W) - q(W)}{1 + q(W)} = A_{\cos}^0(W) - \frac{(1 + A_{\cos}^0(W))q(W)}{1 + q(W)}.$$

Because the denominator $1 + q(W)$ is always greater than or equal to 1, dropping it yields a conservative, linear lower bound. Re-substituting our definition of $q(W)$ brings us to our final operational inequality:

$$A_{\cos}(W) \geq A_{\cos}^0(W) - (1 + A_{\cos}^0(W))\frac{d_{\text{stat}}(W)}{\gamma\|W\|_F}.$$

C.5.5. ANALYSIS

Ultimately, these equations formalize a robust, predictable population mechanism. Weight decay acts as a steering force, continuously selecting for the balance direction $\gamma W$. Because the population Hebbian update naturally maintains a positive alignment with the current weights $W$, any learning signal passing sufficiently close to this balance trajectory is dragged into positive Frobenius and cosine alignment with the Hebbian update.

The alignment analyzed in this work thus degrades predictably with distance from homeostasis. This degradation scales linearly with the absolute distance $d_{\text{stat}}(W)$ for Frobenius alignment, and linearly with the relative distance $d_{\text{stat}}(W)/(\gamma\|W\|_F)$ for cosine alignment.

**C.6. RandomNN formulation**

The RandomNN was a MLP with 3 hidden vectors of size 128 and tanh activations. The MLP took the same input as the student model but outputted a vector of length 4. The output was averaged across the batch and then multiplied by a random projection matrix unique to each parameter and reshaped to be the dimensions of that parameter. No parameters of RandomNN change after initialization. The resulting learning signal for $W$ is a deterministically random low-rank matrix, $W^*$.

The full weight update is given by:

$$\Delta W = \eta\left(g(x,\theta) - \gamma W\right)$$

where

$$g(x,\theta) = p(W^*)s_{dir}(W)s_{red}(W,W^*)W^*$$

and where,

$$s_{red}(W,W^*) = \text{sign}\left(\|W\|_2 - \|W - W^*\|_2\right)$$

$$s_{dir}(W) = \text{sign}\left(100 - \|W\|_2\right)$$

$$p(W^*) = \begin{cases} 1 & \text{if } \|W^*\|_2 \leq 1 \\ \frac{1}{\|W^*\|_2 + \epsilon} & \text{otherwise} \end{cases}$$

The minimal requirements to have non-zero weights and reach stationarity require $g(x,\theta)$ to be some forcing function that wants to make the weights larger than zero. This is the case with any descent learning algorithm, as with zero weights, one can not learn or express anything besides 0. However, it is not only true of learning algorithms.

There are a number of trivial constructions that satisfy this condition, such as setting $f(x,\theta) = A$ where $A$ is a random matrix defined at initialization. This will naturally be an expanding force and become aligned with the Hebbian rule; however, it will do this even without regularization. But is there a way to make a non-learning model that does not behave Hebbian at all without regularization, but does with regularization?

RandomNN is one such construction. In it, we produce random weight update vectors in a subspace of the possible directions of the student model's weight updates. This means that after some number of updates, the value of the weight is not orthogonal to the random update vectors, and in fact becomes highly aligned to them. Thus, for a given weight update, the norm of the weights will either increase or decrease, not strictly increase. We can make an attractor to push the norm of the

weights to a specific non-zero value by choosing to either add or subtract the random update, depending on which one will move it closer to the target value. Thus, without any regularization, the model's weights will converge to have the target norm and will, on average, not increase or decrease, resulting in no Hebbian alignment. However, once a weight decay term is added, the attractor will try to increase to approach the target, and thus align with the Hebbian update. We also apply a weight update norm clip for stability.

