# OpenReview forum: "Ubiquity of Emergent Hebbian Dynamics in Regularized Learning"
_ICML.cc/2026/Conference — ICML 2026 regular_

### Official Review · Reviewer_NH67 · 2026-03-10

**Soundness:** 3
**Presentation:** 3
**Significance:** 3
**Originality:** 3
**Overall Recommendation:** 5
**Confidence:** 3

**Summary:**

The authors provide a mathematical definition of Hebbian alignment and try to answer the question if Hebbian and anti-hebbian weight updates can in fact arise from global error-driven optimization.
They show mathematically and experimentally that near stationarity the degree to which a learning rule prescribes Hebbian aligned (expansive) updates increases with increasing weight decay. They also show that strong noise leads to anti-Hebbian updates.

**Compliance With Llm Reviewing Policy:**

Affirmed.

**Final Justification:**

The authors identify and challenge the belief - that Hebbian and anti-Hebbian weight updates can not arise from global end-to-end optimization - and show that both signatures can arise from variants of backprop based training. My initial concern was that this "widely held belief" sounded like a straw man argument and that the theoretical predictions at equilibrium were somewhat tautological. The reviewers have convinced me that the conclusions can be extended to the non-equilibrium case and that the "widely held belief" is indeed not a straw man argument.

The paper makes compelling predictions backed up by convincing experiments. I am maintaining a score of 5 but raising my confidence from 2 to 3.

There is a remaining issue of confusing terminology regarding gradient estimators and optimizers (see my rebuttal acknowledgement), but I don't think this seriously impacts the main points of the paper.

**Key Questions For Authors:**

- 1. Is it possible to disentangle biological plasticity into “learning” and “decay” terms? The experiment you propose on line 347 seems to hinge on this assumption. I would expect that can only measure the sum, which means that at stationary points where your analysis is valid we would measure approximately nothing.
- 2. In equation 2 and the following paragraph I think your reasoning requires h_a and h_b to be non-negative. Is that correct? The preactivation h_b can attain negative values, and in the case of a tanh activation function, then h_a can also attain negative values.
- 3. Why not compare to contrastive Hebbian learning or Equilibrium propagation? In those frameworks the gradient estimates are compose of a Hebbian and an anti Hebbian term, which I would think would make the analysis easier for you.
- 4. Did you use Adam or AdamW? Do you think this may have influenced your results?
- 5. Is it really a widely held belief that Hebbian and anti-Hebbian weight updates can not arise from global end-to-end optimization? It sounds a bit like a straw man argument, but this is outside of my domain.

**Limitations:**

The impact statement is sufficient.

**Strengths And Weaknesses:**

## Strengths
- You identify and challenge the belief - that Hebbian and anti-Hebbian weight updates can not arise from global end-to-end optimization - and show that both signatures can arise from variants of backprop based training.
- The analysis makes very clear predictions, which are experimentally confirmed. Figure 4 in particular is very convincing!
- The analysis hints at ways to test parts of the theory in biological systems (line 347 column 1). Strong noise should correlate with anti-Hebbian plasticity.

## Weaknesses
- Disentangling the weight updates into the sgd term (the gradient of the loss with respect to the weights) and the regularization term is straight-forward in digital simulations, but in a biological system I expect they would be difficult to disentangle.
- Juxtaposing SGD, ADAM and DFA does not really make sense.
SGD, ADAM, and other common optimizers tell you how to update weights given a gradient estimate. You can apply them to gradient estimates computed by backprop, DFA or any other learning algorithm you might like.
Backpropagation and direct feedback alignment are algorithms used to compute gradients (or in the case of DFA a surrogate gradient).
So it would make sense to also consider DFA with different optimizers (since you do that for backpropagation).

## Other comments
- ADAM is probably a bad choice for your experiments as it will compute its adaptive learning rates based on the gradient of the sum of the loss and the weight decay term, which is almost certainly not what you want here. You should probably be using AdamW, which only uses the loss to compute its adaptive learning rates. This may be why your experiments with strong weight decay failed for ADAM in table 1.
- Figure 19: The y-axis limits are too large. You can probably narrow it down to [-0.15, 0.15], to make the plot readable.
- Line 71, column 2: The acronym BCM is not defined anywhere.
- Figure 2 needs a colorbar.
- The observation that at fixed points weight decay leads to Hebbian gradients is almost tautological. If you have two forces and one is contractive, then by definition at equilibrium the other one must be expansive.

---

> ### Author Rebuttal · Authors · 2026-03-31
>
> Dear reviewer **NH67,** thank you for your careful review of our manuscript.
>
> *W1: Disentangling the weight updates... in a biological system I expect they would be difficult to disentangle. Q1: Is it possible to disentangle biological plasticity into “learning” and “decay” terms? ....*
>
> Thanks for raising this point. It is likely possible to separate these two mechanisms since they probably operate on different time scales and might be measured in LTP experiments. We provide a discussion of this in the “Possible Neurobiological Mechanism” section on line 203.  Additionally, we believe that our theory has the potential to inspire and open up new experimental tests of biological plasticity. It is common for theoretical models to precede experimental tools; indeed, Hebbian learning was proposed 20 years before it could be experimentally measured. We will expand our text to clarify this context.
>
> *W2: Juxtaposing SGD, ADAM and DFA does not really make sense...*
>
> These are deep networks, so DFA is not a direct gradient estimator. Adam and SGD both are, but do different things with the gradients, so they are interesting to look at. Using DFA with optimizers is fairly non-standard, so we did not include experiments for that.
>
> *W3: ADAM is probably a bad choice... may be why your experiments with strong weight decay failed for ADAM in table 1. Q4: ... Do you think this may have influenced your results?*
>
> Thanks for pointing this out! We did use normal Adam for these experiments with weight decay. While it's possible this may have contributed to the weight collapse at high weight decays, we don't think this materially influenced the trends.
>
> *W4: Figure 19: The y-axis limits are too large. You can probably narrow it down to [-0.15, 0.15], to make the plot readable.*
>
> We will change the Y axis of the plot, but the main trend is that the alignment is around zero at steady state.
>
> *W5: Line 71, column 2: The acronym BCM is not defined anywhere.*
>
> We have added this to the paper; it stands for "Bienenstock-Cooper-Munro." We added this to the introduction.
>
> *W6: Figure 2 needs a colorbar.*
>
> The visualization shows the gradients divided by their norm, and the colors are from the min and max of the respective source. Red is positive, blue is negative. The point of this plot is to show the alignment in direction, not magnitude. We have updated the description of the figure to clarify this.
>
> *W7: The observation that at fixed points weight decay leads to Hebbian gradients is almost tautological. If you have two forces and one is contractive, then by definition at equilibrium the other one must be expansive.*
>
> The weight decay theory presented in the body of the work is very elegant, though it does require substantially more work to show that it holds outside of exact homeostasis. We provide a proof in Appendix of this 5.C and also provide support through extensive empirical experiments, including Figures 11, 15, and 16 of the text.
>
> Q2: In equation 2 and the following paragraph I think your reasoning requires h_a and h_b to be non-negative. Is that correct? The preactivation h_b can attain negative values, and in the case of a tanh activation function, then h_a can also attain negative values.
>
> We do not require $h_a$ or $h_b$ to be positive, only that $h_a h_a^\top$ is positive-semidefinite, which is true for any real valued $h_a$. We actually see very strong alignment emprically for tanh activations (see Figure 3).
>
> Q3: Why not compare to contrastive Hebbian learning or Equilibrium propagation? In those frameworks, the gradient estimates are compose of a Hebbian and an anti Hebbian term, which I would think would make the analysis easier for you.
>
> Thanks for raising this question. In Contrastive Hebbian learning, for example, the Hebbian term is mechanistic and not emergent. Our focus is also algorithm-agnostic (instead of on specific rules such as EP or CHL). This is why we compared across different learning rules (such as SGD, Adam, DFA, etc). Similar statements hold for equilibrium propagation.
>
> Q5: Is it really a widely held belief that Hebbian and anti-Hebbian weight updates can not arise from global end-to-end optimization? It sounds a bit like a straw man argument, but this is outside of my domain.
>
> It is widely held and not a straw man argument. Hebbian and anti-Hebbian learning are entirely local learning rules since they are homo-synaptic, so alone they are not capable of doing global optimization. There are some attempts to make Hebbian-esque learning rules that are capable of SGD and thus global optimization, which we address in the "Gradient Descent in the Brain" and "Similarity between learning algorithms" background sections on lines 87 and 102.

---

> > ### Author Rebuttal · Reviewer_NH67 · 2026-04-03
> >
> > Thank you for the detailed answers. I am maintaining the same score but raising my confidence from 2 to 3.
> >
> > Regarding ADAM, SGD and DFA, I think it is quite unfortunate to group them all together in the category of "learning rules". I encourage you to make a distinction between gradient estimators and optimizers.
> >
> > Backprop is a gradient estimator, DFA is a surrogate/pseudo gradient estimator (due to the heuristic modification of the backward path). Optimizers like ADAM and SGD are rules that define how you update the weights given a gradient estimate (or a surrogate thereof). To make this point clear: [Nøkland 2016](https://arxiv.org/pdf/1609.01596) used the RMSProp optimizer in conjunction with DFA and [Moskovitz et al 2019](https://arxiv.org/pdf/1812.06488) used ADAM in conjunction with DFA. So what you are comparing is not SGD, ADAM and DFA. What you are comparing is backprop+SGD, backprop+ADAM and DFA+SGD.

---

> > > ### Author Response · Authors · 2026-04-07
> > >
> > > Thank you very much for your review of our work. We will make this clarification in the final manuscript.

---

### Official Review · Reviewer_4Vkr · 2026-03-13

**Soundness:** 2
**Presentation:** 3
**Significance:** 3
**Originality:** 3
**Overall Recommendation:** 3
**Confidence:** 4

**Summary:**

The paper studies whether Hebbian and anti-Hebbian signatures observed in synaptic updates must reflect genuinely Hebbian learning mechanisms, or whether they can also arise emergently from more general optimization dynamics. The main thesis is that, near stationarity, L2 weight decay pushes the learning-signal component of updates toward positive alignment with a Hebbian direction, while sufficiently strong stochastic perturbations can instead induce anti-Hebbian alignment. The paper presents both a theoretical argument and empirical experiments across several architectures and update rules.

On the theory side, the paper derives positive Hebbian alignment at stationarity under a chain-rule structure and a strong norm-decoupling assumption, and extends this with local bounds near stationary points and in terms of a stationarity-gap quantity.   On the empirical side, it evaluates classification and student-teacher regression settings using MLPs and transformers, and reports that Hebbian alignment increases with weight decay across SGD, Adam, DFA, and even a synthetic RandomNN update construction; it also studies a noise-vs-weight-decay phase transition for anti-Hebbian behavior.

Overall, the paper aims to reinterpret Hebbian-looking plasticity as potentially emergent rather than uniquely mechanistic, with implications for both neuroscience interpretation and optimization theory.

**Compliance With Llm Reviewing Policy:**

Affirmed.

**Final Justification:**

The rebuttal clarified several useful points, especially the intended scientific role of the RandomNN experiment, the separation between the ANN optimization result and the broader neuroscience interpretation, and the authors' willingness to further tighten the presentation. I agree that the paper is strongest as an optimization/learning-dynamics contribution, and I find the central question both interesting and original.

However, my main reservations remain. The core theoretical results still rely on fairly strong assumptions, and the broader neuroscience interpretation remains more tentative than the optimization claim itself. While the ANN experiments support the qualitative phenomenon well, I still think the paper would benefit from a clearer distinction between what is formally proved, what is demonstrated empirically in artificial neural networks, and what is offered as a broader interpretive implication for biological plasticity.

Overall, I appreciate the paper's conceptual ambition and the helpful rebuttal, but I do not see enough change to my overall assessment. I am therefore keeping my score and recommendation unchanged.

**Key Questions For Authors:**

1. How robust are the theoretical claims to approximate rather than exact norm decoupling?
2. How general is the phenomenon in larger modern architectures?
3. What is the precise scientific role of the RandomNN experiment?
4. Can the authors better distinguish the optimization claim from the neuroscience claim?

**Limitations:**

No. The authors do discuss several technical limitations, including focus on smaller models and incomplete understanding of anti-Hebbian regions in larger settings, which is good.  However, the paper should more clearly delimit which conclusions are about ANN optimization experiments versus neuroscience interpretation.

**Strengths And Weaknesses:**

### Strengths

1. Interesting and important question.
The identifiability question is well posed: the paper asks whether observing Hebbian or anti-Hebbian update structure is sufficient evidence for a Hebbian learning mechanism, or whether similar signatures can emerge from regularized gradient-based dynamics. This is a meaningful conceptual contribution at the neuroscience/ML interface.

2. Clear central mechanism.
The core argument tying stationarity, weight decay, and alignment to Hebbian directions is intuitive and mathematically clean. The main derivation in the body of the paper is straightforward, and the appendix provides a more formal version with local bounds near stationary points.

3. Broad empirical coverage within the chosen scope.
The experiments span classification and regression, MLPs and transformers, several optimizers, and several regularization/noise settings. The tables and figures consistently support the qualitative trend that stronger weight decay increases Hebbian alignment, while stronger noise can induce anti-Hebbian alignment.

4. The paper is honest about some limitations.
The authors explicitly note that they mainly study smaller models and that the stationarity-based explanation does not yet fully account for all anti-Hebbian regions observed in larger models. This is a useful limitation statement.

### Weaknesses

1. The main theoretical assumptions are quite strong.
The formal results rely on assumptions such as exact norm decoupling of presynaptic activations and local boundedness/Lipschitz continuity around a nonzero stationary point. These assumptions may be reasonable in normalized or collapse-like regimes, but they are restrictive, and the paper does not sufficiently clarify when they should be expected to hold in realistic deep networks.

2. The empirical evidence is suggestive rather than decisive for the broader neuroscience claims.
The experiments are entirely in artificial neural-network settings, mostly relatively small-scale. While the optimization phenomenon itself is plausible, the stronger interpretive claim about biological plasticity remains speculative without any closer bridge to neural data or biologically grounded models beyond analogy.

3. Some claims are broader than the evidence supports.
The manuscript frequently uses language like "ubiquity” and frames the phenomenon as broadly general, but it also acknowledges that in larger or different settings the behavior can become more mixed and the theory less predictive. I would prefer more careful separation between what is proven, what is empirically observed in toy/small experiments, and what is conjectural.

4. Relevance of the RandomNN experiment is mixed.
The RandomNN construction is useful for making the point that a Hebbian-looking signature need not imply meaningful learning, but it also feels somewhat artificial. The paper would benefit from a clearer explanation of why this construction strengthens the scientific claim rather than just illustrating a formal possibility.



### Overall

Soundness: The theoretical argument is coherent, but rests on strong assumptions and is only partially matched to realistic deep-learning regimes. The experiments support the qualitative trend, though not the strongest framing.

Presentation: The paper is generally readable and structured well, with a clear story from mechanism to experiments to discussion.

Significance: The question is important and the results may be useful for both interpretability of learning dynamics and the neuroscience/ML dialogue.

Originality: The central reframing of Hebbian signatures as emergent consequences of regularization/noise balance is novel and thought-provoking.

---

> ### Author Rebuttal · Authors · 2026-03-31
>
> Thank you for the review. Think there may have been some ambiguity in our presentation, so below we have indicated where a number of these issues are addressed in the manuscript. We would be grateful for any additional, specific clarifications you could provide to help us improve the paper.
>
> *W1: The main theoretical assumptions are quite strong. The formal results rely on assumptions such as exact norm decoupling of presynaptic activations and local boundedness/Lipschitz....*
>
> We agree that the main theory presented in the body of the paper is under fairly idealized conditions in order to make the proof elegant and easy to follow; however, we relax many assumptions and provide a rigorous proof in Section 5.C of the Appendix. Additionally, we provide extensive empirical evidence throughout the paper, including a random learning rule that still exhibits this behavior far from stationarity (see Figures 11, 15, and 16).
>
> *W2: The empirical evidence is suggestive rather than decisive...*
>
> Thanks for this criticism. Our work is theoretical and computational in nature, and it is true that we do not study actual biological experiments, and we believe that our theory has the potential to inspire and open up new experimental tests of biological plasticity. As is clear from the manuscript, one of our key contributions is a new mechanism (the “emergent Hebbian”) for Hebbian-like plasticity, and we did provide a section dedicated to this point. See “Possible Neurobiological Mechanism” in line 203. A primary goal of our theoretical framework is to draw attention to this mechanism so that the experimental neuroscience community can help design targeted tests. It is common for theoretical models to precede experimental tools; indeed, Hebbian learning was proposed 20 years before it could be experimentally measured. We will expand our text to clarify this context.
>
> *W3: Some claims are broader than the evidence supports. The manuscript frequently uses language like "ubiquity”....*
>
> We agree that the use of “ubiquity” is important to qualify well, though we respectfully disagree that the paper overclaims its core result. Our main claims are clearly scoped to the near-homeostatic regime, and in the manuscript, we distinguish between (i) what is proved theoretically, (ii) what is supported empirically across architectures and learning rules, and (iii) the broader interpretive implications for neuroscience. That said, we are happy to further tighten wording wherever the current presentation may have suggested a broader claim than intended. In the current draft, all of our claims of ubiquity in the title, abstract, and conclusion are qualified as being near homeostasis, with a more comprehensive proof of this in Appendix 5.C. That said, we could be clearer when using the term “universal” when discussing the Random NN regression, though we have struggled to find other ambiguous instances. We don’t provide any conjectures in this work, and all theorems presented are also supported by empirical evidence. If there is a specific sentence or claim that still reads as overly broad to the reviewer, we would be grateful for the exact quotation so we can revise it directly.
>
> *W4: Relevance of the RandomNN experiment is mixed. The RandomNN construction is useful for making the point that a Hebbian-looking signature need not imply meaningful learning, but it also feels somewhat artificial. The paper would benefit from a clearer explanation of why this construction strengthens the scientific claim rather than just illustrating a formal possibility. Q3: What is the precise scientific role of the RandomNN experiment?*
>
> The scientific role of the RandomNN construction is to demonstrate that any stable learning rule paired with weight decay produces this trend. It proves our core thesis that a Hebbian-looking signature does not strictly require a mechanistically Hebbian rule. This is the most general form of empirical support for the optimization dynamics we describe. We will further detail this rationale in Appendix C.6.
>
> *Q1: How robust are the theoretical claims to approximate rather than exact norm decoupling?*
>
> The more general part of the theoretical claim in Eq. (13)-(14) are robust and does not require the decoupling assumption.
>
> Q2: How general is the phenomenon in larger modern architectures?
>
> We have verified that this phenomenon scales. We ran this model on ResNet-18, ResNet-34, and ResNet-50 and observed a very strong alignment of approximately 0.8, both at convergence and well before.
>
> *Q4: Can the authors better distinguish the optimization claim from the neuroscience claim?*
>
> We believe the current manuscript already structurally separates these points, particularly in the "Possible Neurobiological Mechanism" section on line 203. We are unsure which specific passages the reviewer feels conflate the two. If the reviewer could point to the precise section or claim they have in mind, we would be happy to clarify the text.

---

> > ### Author Rebuttal · Reviewer_4Vkr · 2026-04-02
> >
> > The authors clarified some important points, especially  the role of the RandomNN experiment, and the intended separation between the optimization result and the neuroscience interpretation.
> >
> > To me, the paper is strongest on the ANN optimization side, while the neuroscience interpretation still feels more tentative. I also think the main theory relies on fairly strong assumptions.
> > So I consider my concerns partially resolved, and I will keep my score unchanged.
> >
> > My follow-up question is: can the revised paper make even clearer what is proved, what is shown in ANN experiments, and what is meant as broader neuroscience interpretation?

---

> > > ### Author Response · Authors · 2026-04-03
> > >
> > > **4Vkr's acknowledgement:** (c) Partially resolved or unresolved, but the remaining concerns … requires a significant update to the paper.
> > >
> > > **What 4Vkr believes are some of the strengths:**
> > >
> > > S1: Motivating the role of the RandomNN experiment
> > >
> > > S2: The intended separation between the optimization result and the neuroscience interpretation.
> > >
> > > S3: Paper is strongest on the ANN optimization
> > >
> > > **What 4Vkr believes are some of the remaining weaknesses:**
> > >
> > > W5: The neuroscience interpretation still feels more tentative.
> > >
> > > W6: I also think the main theory relies on fairly strong assumptions.
> > >
> > > Q5: Can the revised paper make even clearer what is proved, what is shown in ANN experiments, and what is meant as broader neuroscience interpretation?
> > >
> > >
> > >
> > > We thank you for your review of our rebuttal and acknowledgement of S1, S2, and S3. We are glad you feel we have sufficiently answered Q5 in our rebuttal, as per S2. In the revision we will add the **two additional sentences** we mentioned in our rebuttal that represent an idea that was not present in the original paper: “A primary goal of our theoretical framework is to draw attention to this mechanism so that the experimental neuroscience community can help design targeted tests. It is common for theoretical models to precede experimental tools; indeed, Hebbian learning was proposed 20 years before it could be experimentally measured.” Thus, we agree that the neuroscience instantiation of this effect is currently a bit more tentative (W5), though again, we view this as a reason this paper should be published in order to get the wider community to think about this challenging question.
> > >
> > > We agree that the theory presented in the body of the work is an illustrative theory analyzing a simple setting (W6), though we view this as necessary for the work to be broadly understood across different fields, such as AI and neuroscience, in which **readers may not have the sufficient background to understand** the more thorough **5-page derivation in which we relax many assumptions that we present in Appendix 5.C.**

---

### Official Review · Reviewer_8Ldh · 2026-03-13

**Soundness:** 3
**Presentation:** 3
**Significance:** 2
**Originality:** 3
**Overall Recommendation:** 5
**Confidence:** 3

**Summary:**

The paper proposes the idea that phenomenologically Hebbian and anti-Hebbian plasticity can emerge as byproduct of much more general learning rules (gradient based) that include weight decay (a contractive force) and/or noise (an expansive force).

The authors postulate that gradient descent training with weight decay (especially with high weight decay values) results into weight updates that are similar to those happening under purely Hebbian learning. They offer a clear argument that since weight decay is contractive, the learned gradient part of the update must be expansive on average, and an expansive, rank-1-looking update will look Hebbian. The authors propose an interesting explanation for why Hebbian-looking updates can show up during learning with weight decay even when the underlying algorithm is not Hebbian. They find that **stronger weight decay, larger learning rate, and larger batch size** lead to better alignment between gradient-based and Hebbian weight updates. Moreover they find that **strong noise** in learning results in a learning signal that is anti-Hebbian. The authors mention that when noise co-exists with weight decay there is a competition between the two forces that contribute Hebbian and anti-Hebbian aligned updates, and they identify a “phase transition” when the interplay between these two forces changes polarity.

Overall the authors put forward the argument that Hebbian and anti-Hebbian plasticity might be a byproduct or components of a more general gradient-like weight optimisation that might co-exist with purely local Hebbian updates. While I find it an interesting and thought provoking argument, I am not convinced by their empirical justification since the approach mostly holds for stationary states.

While the authors mention that their theory motivates experiments that aim to disambiguate between purely Hebbian synaptic updates vs. emergent Hebbian-like synaptic updates that are byproducts of gradient-descent like objectives, they neither provide suggestions on what type of experiments could disambiguate these two cases, neither provide convincing enough empirical justification that would lead to motivate experimentalists to devise such experiments.

**Compliance With Llm Reviewing Policy:**

Affirmed.

**Final Justification:**

The paper is well written, proposes a fairly interesting idea - that seemingly (anti)Hebbian updates may show up during learning with weight decay even when the underlying mechanism is not Hebbian - presented clearly and demonstrated thoroughly on artificial neural networks.

My main concerns raised during the review were resolved and thus now I am in full support of the paper!

**Key Questions For Authors:**

- I recently became aware of the notion of "Gradient dissent" (not a typo) in LLM, where the cross-entropy gradient is decomposed into an attractive component that increases the correct logit and a repulsive component that decreases incorrect logits, and during training these components become  systematically opposed,
for instance here [here](https://openreview.net/forum?id=c1pEF21dsY). I see a relation between the claims of this paper and the one linked that the training in both systems stalls because of structured opposition of the update components. Not sure if any reasonable connection can come out of it, but I thought the authors would find this part of the machine learning literature probably interesting.

- Can the authors demonstrate a more direct relationship between the observed Hebbian-like alignment in their framework and known neurophysiological signatures of Hebbian plasticity? Or propose experiments how to delienate purelly hebbian signatures from hebbian-like signatures of plasticity?

- Related to some comments I mentioned above, I think that the paper would substantially improve if the authors would attempt to connect their framework to existing theories of Hebbian plasticity and homeostatic mechanisms.

- I would expect the authors to connect the proposed framework with somewhat related literature that postulates that  similarity-matching objectives can produce explicit Hebbian feedforward + anti-Hebbian lateral local learning rules, e.g. see [1] (although in the present paper the authors consider a pure feed forward structure I think a discussion would be insightful). The two papers are somewhat oppositte in their explanatory direction, but more or less associate local (anti) Hebbian learning rules to global optimization objectives, thus considering this paper and related papers from the same authors render the claim "Hebbian rules emerge from global optimization objectives" less novel. But I would be interested to hear the perspective of the authors on that topic.

- See also the weaknesses.

## Minor

- In line 74, first column, I think what is meant is "strengthening with increasing decay"

- I understand that the claim the authors attempt to make is that global optimization objectives may appear as locally Hebbian/anti-Hebbian updates, do you think there would be a way to employ your framework to work towards the opposite direction, on how one could employ or coordinate local learning rules to optimize a global optimization objective?

- Other related papers the authors might want to discuss: [2]

-----

## References

[1] Pehlevan, C., Sengupta, A. M., & Chklovskii, D. B. (2017). Why do similarity matching objectives lead to Hebbian/anti-Hebbian networks?. Neural computation, 30(1), 84-124.

[2] Zenke, F., Gerstner, W., & Ganguli, S. (2017). The temporal paradox of Hebbian learning and homeostatic plasticity. Current opinion in neurobiology, 43, 166-176.

**Limitations:**

- The main limitation is that the authors over-claim the ubiquity of their of their proposed framework while the theoretical part clearly only holds for the stationary case.

**Strengths And Weaknesses:**

## Strengths

- The proposed idea is original and thought provoking.
-  The authors have performed an extensive number of numerical experiments where they explore the alignment of the weight updated with the Hebbian weight updates for different parameter values of weight decay, noise amplitude, network sparsity, network size, learning rate, batch size, and for different learning rules (namely Adam, stochastic gradient descent, direct feedback alignment, and randomNN) and regularizers.

 -   The authors explicitly demonstrate that is the gradient part of the weight update that aligns with the Hebbian updates and not the full weight update, and thus there is no issue with weights growing infinitely large (since there is also a weight decay).

 -   The authors propose an interesting theory: gradients under realistic constraints project onto Hebbian-looking directions, and thus observing Hebbian plasticity in experiments does not rule out gradient-like learning.







-----

## Weaknesses

- There is insufficient support for the strongest claims of the paper.
- The biological relevance of the main mechanism is insufficiently established. It is unclear to me according to the paper how or whether the roughly Hebbian-aligned behavior identified in the paper is actually related to the kinds of effects measured in neurophysiological Hebbian plasticity experiments.
- The authors provide extensive experimental validation of their claims in the main text and in the appendix, however, as I understand, the proposed effect is evident only for fairly small architectures and for mostly stationary settings.
- The link to experimental validation is underdeveloped.
- Insufficiently justified claim about "universality".
- As I understand the effect is observable only in small networks, with the authors mentioning that they struggled to observe it in large networks, and in networks with no recurrence.

---

> ### Author Rebuttal · Authors · 2026-03-31
>
> Dear reviewer **8Ldh,** thank you for your review of our manuscript. We provide responses to your questions below. Many of your questions touched on the same core points so we have grouped them for clarity.
>
> *W2: ...roughly Hebbian-aligned behavior identified in the paper is actually related to the kinds of effects measured in neurophysiological Hebbian plasticity experiments. W4: The link to experimental validation is underdeveloped. Q2: Can the authors demonstrate a more direct relationship ... or propose experiments...?*
>
> We thank the reviewer for highlighting this. We do discuss a “Possible Neurobiological Mechanism” (line 203) through LTP experimentation, though these signals could be implemented in a different way that is more difficult to differentiate. A primary goal of our theoretical framework is to draw attention to this mechanism so that the experimental neuroscience community can help design targeted tests. It is common for theoretical models to precede experimental tools; indeed, Hebbian learning was proposed 20 years before it could be experimentally measured. We will expand our text to clarify this context.
>
> *W3: The authors provide extensive experimental validation ...  the proposed effect is evident only for fairly small architectures and for mostly stationary settings. W6: ...authors mentioning that they struggled to observe it in large networks, and in networks with no recurrence.*
>
> We appreciate your criticism. As shown in Figure 9, the effect can become even stronger with model size. We test models up to 50 million parameters, which is the same order of magnitude as the number of synapses in a fly’s brain. We hope that in the future, a researcher with access to more computing power will test this on even larger architectures, but from our perspective, a 50M is sufficient to demonstrate this is a real trend that is worth further research. We also provide a proof in Appendix 5.C that demonstrates this behavior out of stationarity as well as empirical support in Figures 11, 15, and 16. We also note that the experiments in the manuscript were conducted on non-recurrent architectures.
>
> *W1: ...insufficient support for the strongest claims... W5: Insufficiently justified claim about "universality". Q8: The main limitation is that the authors over-claim the ubiquity of their of their proposed framework while the theoretical part clearly only holds for the stationary case.*
>
> Thanks for this criticism. We will work to further qualify in the revision. We do specify ubiquity as applying specifically *near* homeostasis in the title, abstract, and conclusion, though we will also clarify this in the Random NN Regression paragraph. We also note that our results do indeed apply outside of perfect stationarity; we provide a lengthy proof of this in Appendix 5C and show example runs in Figures 11, 15, and 16 in which the models exhibit alignment far before convergence.
>
>
>
>
> *Q1: I recently became aware of the notion of "Gradient dissent".... Q3: ...authors would attempt to connect their framework to existing theories of Hebbian plasticity and homeostatic mechanisms. Q4: ...similarity-matching objectives can produce explicit Hebbian feedforward + anti-Hebbian lateral local learning rules, e.g. see [1]...  Q7: Other related papers the authors might want to discuss: [2]*
>
> We thank the reviewer for these references and will add them to our discussion. We do provide a fairly comprehensive account of existing theories of the Hebbian literature in Section 2 (line 98), and specifically the final paragraph of the “Hebbian Learning” section on line 76. But you have pointed us towards some really interesting other papers that we will be sure to address. Gradient dissent is very interesting, though we aren't sure exactly how to connect it. It is well established that Hebbian and anti-Hebbian structures can arise in specific settings. Pehlevan et al. and Zenke et al. provide two more examples of how it can emerge from optimizing when similarity min-max objectives, or more structurally from stability requirements. But rather than deriving Hebbian/anti-Hebbian rules from particular objectives or stability requirements under Hebbian assumptions, we show that Hebbian-like alignment can emerge near homeostasis regardless of the learning rule or the learning objective used. Thus, our claim is more general.
>
> *Q5: In line 74, first column, I think what is meant is "strengthening with increasing decay"*
>
> Thanks! We will fix this.
>
> *Q6: ...how one could employ or coordinate local learning rules to optimize a global optimization objective?*
>
> This is an important and open problem. In some cases, Hebbian learning does behave like gradient descent (such as in linear autoencoders), but it is unclear to what extent this is generalizable.

---

> > ### Author Rebuttal · Reviewer_8Ldh · 2026-04-03
> >
> > I thank the authors for the honest rebuttal and their engagement with my and the other reviewers' comments. Most of my concerns have been resolved on the condition that the authors will make the mentioned updates in the camera ready paper. I will update my score.

---

> > > ### Author Response · Authors · 2026-04-07
> > >
> > > Thank you very much for your review of our work. We will make the proposed revisions to the final manuscript.

---

### Official Review · Reviewer_zs7V · 2026-03-20

**Soundness:** 4
**Presentation:** 3
**Significance:** 3
**Originality:** 3
**Overall Recommendation:** 5
**Confidence:** 3

**Summary:**

The paper shows that Hebbian-like synaptic updates occurs during supervised training near equilibrium (when weight changes are small), and anti-Hebbian updates are driven by noise. This suggests caution when interpreting Hebbian computation, as it can arise from different mechanisms.

**Compliance With Llm Reviewing Policy:**

Affirmed.

**Final Justification:**

I thank the authors for the reply -- my concerns and questions have been addressed. I will maintain my score.

**Key Questions For Authors:**

1. Can you give a potential explanation for why solutions with high generalization had low Hebbian and anti-Hebbian alignment? As well why does the validation loss get smaller with smaller weight decay and noise? Is that related to details of this experiment?
2. Can you explain the condition in 158? Should h_a depend on x, and so how does the statement depend on x?
3. Minor: should you have a factor of 2 in equation in equation 16 (or 15)?

**Limitations:**

Yes

**Strengths And Weaknesses:**

Strengths:
- overall well written paper, the figures were clear, and overall the message was clear (though see comment on introduction)
- the authors make a clear point about how Hebbian and non-Hebbian like updates occurs due to the weight decay and noise
- the claims are supported by the derivations in the main paper, and the experiments are well designed and support the main claims

Weaknesses (mostly minor):
- Writing is far too focused from the first paragraph. For example, the first paragraph uses terms like “homeostatic” and “homosynaptic”, Hebbian, as well as acronym STDP (not defined). I think the first paragraph could be written to appeal to a broader audience.
- L48, right column: sentence doesn’t make sense “; constraints, while near homeostasis”
- Line 70: define BCM
- See also the questions

---

> ### Author Rebuttal · Authors · 2026-03-31
>
> Dear reviewer **zs7V**, thank you for your careful review of our manuscript. We provide responses to your questions below.
>
> *W1: Writing is far too focused from the first paragraph. For example, the first paragraph uses terms like “homeostatic” and “homosynaptic”, Hebbian, as well as acronym STDP (not defined). I think the first paragraph could be written to appeal to a broader audience.*
>
> We have updated the document to revise all of your minor weaknesses. Our new introductory paragraph is:
>
> > A central question in computational neuroscience is the extent to which artificial intelligence systems capture the computational motifs found in the human brain. Traditionally, it has been widely believed that artificial intelligence and the human brain operate under entirely different learning mechanisms: modern AI systems rely on gradient descent for learning, whereas biological synapses are generally thought to adapt primarily via correlation-based mechanisms, specifically Hebbian and anti-Hebbian forms of learning (Koch et al., 2013; Zenke & Gerstner, 2017; Lisman, 1989; Lamsa et al., 2007; Abbott & Nelson, 2000; Magee & Grienberger, 2020). Standard Hebbian theory views Hebbian learning as mechanistically implemented by local pre/post-activation homo-synaptic rules such as Spike-Time-Dependent Plasticity (STDP). This Hebbian update is widely believed to be the primary driver of learning and memory in the brain, with additional balancing mechanisms preventing runaway neuronal growth or decay (Oja, 1982b; Bienenstock et al., 1982; Caporale & Dan, 2008; Froemke et al., 2005; Brzosko et al., 2019; Zenke & Gerstner, 2017). This framework has been extraordinarily successful in explaining how local plasticity can extract structure from inputs and why homeostatic constraints are essential.
> > At the same time, Hebbian-like signatures are often used as indirect evidence against global error-driven optimiza-
> tion in the brain, since gradient-based learning is typically viewed as requiring nonlocal error signals and coordination
> that neural circuits may not provide (Hebb, 2005; Rumelhart et al., 1986; Whittington & Bogacz, 2019; Lillicrap et al., 2020). This motivates a basic identifiability question: does observing Hebbian/anti-Hebbian structure in updates uniquely imply an underlying Hebbian computation?
>
>
> *W2: L48, right column: sentence doesn’t make sense “; constraints, while near homeostasis”*
> Thanks. We have revised to:  "…we propose an additional, coexisting account of why synaptic modifications can appear Hebbian or anti-Hebbian: close to a homeostatic regime, regularization constraints project complex learning dynamics into Hebbian directions."
>
> *W3: Line 70: define BCM.*
>
> We have added this to the paper; it stands for “Bienenstock-Cooper-Munro.”
>
> *Q1: Can you give a potential explanation for why solutions with high generalization had low Hebbian and anti-Hebbian alignment? As well why does the validation loss get smaller with smaller weight decay and noise? Is that related to details of this experiment?*
>
> This is a good question that is answered by the proposed mechanism. Significant alignment either implies one has too strong regularization or noise, which impairs the optimization of the problem, leading to worse performance. The validation loss gets smaller with smaller weight decay and noise because weight decay and noise are both regularizers, and when over-constraining the functions that can be learned, it leads to worse solutions.
>
> *Q2:  Can you explain the condition in 158? Should h_a depend on x, and so how does the statement depend on x?*
>
> Yes. Here, $h_a$ is the representation of a given $x$. The assumption here states that the representations of all inputs have an identical norm, and so is a constant. Thus, only the direction of $h_a$ depends on $x$, not its norm.
>
> *Q3: Minor: Should you have a factor of 2 in equation 16 (or 15)?*
>
> Yes. Thanks, we have added it in equation 15.

---

> > ### Author Rebuttal · Reviewer_zs7V · 2026-04-02
> >
> > I thank the authors for the reply -- my concerns and questions have been addressed. I will maintain my score.

---

> > > ### Author Response · Authors · 2026-04-07
> > >
> > > Thank you very much for your review of our work.

---

### Decision · Program_Chairs · 2026-04-30

**Decision:**

Accept (regular)

**Comment:**

Three out of four reviewers suggested acceptance, despite some concerns about the general applicability of the results. It appears that the authors have reworked and clarified a work previously submitted elsewhere. Overall I am leaning towards acceptance.